# Regulation of telomere silencing by the core histones–autophagy–Sir2 axis

Qianyun Mei*, Qi Yu* , Xin Li, Jianguo Chen, Xilan Yu

**Telomeres contain compacted heterochromatin, and genes adjacent to telomeres are subjected to transcription silencing. Maintaining telomere structure integrity and transcription silencing is important to prevent the occurrence of premature aging and aging-related diseases. How telomere silencing is regulated during aging is not well understood. Here, we find that the four core histones are reduced during yeast chronological aging, leading to compromised telomere silencing. Mechanistically, histone loss promotes the nuclear export of Sir2 and its degradation by autophagy. Meanwhile, reducing core histones enhances the autophagy pathway, which further accelerates autophagy-mediated Sir2 degradation. By screening the histone mutant library, we identify eight histone mutants and one histone modification (histone methyltransferase Set1-catalyzed H3K4 trimethylation) that regulate telomere silencing by modulating the core histones–autophagy–Sir2 axis. Overall, our findings reveal core histones and autophagy as causes of aging-coupled loss of telomere silencing and shed light on dynamic regulation of telomere structure during aging.**

## Introduction

Aging is a progressive degenerative state accompanied by loss of physiological integrity, leading to functional decline and vulnerability to death (Lopez-Otin et al, 2013; Chakravarti et al, 2021). Aging is a leading risk factor for the increased incidence of many human diseases, including Alzheimer's disease, cardiovascular disorders, cancer, and so on (Melzer et al, 2020; Chakravarti et al, 2021). One important hallmark of aging is telomere dysfunction (Lopez-Otin et al, 2013). Telomeres have a typical heterochromatin structure, and genes located within or near the telomeres are expressed at a low level or silenced irrespective of their promoter sequences, a phenomenon called telomere position effect, which has been observed in a variety of organisms, including yeast, *Drosophila*, and mammals (Gottschling et al, 1990). Telomere silencing is important

for healthy aging of postmitotic cells, and many proteins involved in transcription silencing also regulate life span (Smeal et al, 1996; Dang et al, 2009; Kozak et al, 2010). Moreover, as some stress-resistant genes are located within subtelomeric regions, telomere silencing is important for cell response to external stimuli (Ai et al, 2002).

In budding yeast, transcription silencing at telomeres requires the interaction between histones and silent information regulator (SIR) complex, which is composed of Sir2, Sir3, and Sir4 (Hecht et al, 1995). Sir2 is a NAD$^+$-dependent histone deacetylase, which specifically deacetylates H4K16. The SIR complex protects the underlying genes from transcription machinery, leading to their transcriptional silencing (Gottschling, 1992). The activity of SIR complex and its association with chromatin are regulated by histone modifications, including H4K16 acetylation, H3K4 trimethylation (H3K4me3), H3K79 trimethylation, H3T11 phosphorylation, and H4K12 acetylation (Santos-Rosa et al, 2004; Dang et al, 2009; Zhou et al, 2011; Xue et al, 2015; Zhang et al, 2021; He et al, 2022). H3K4me3 is catalyzed by histone methyltransferase Set1, which is required to maintain telomere silencing (Nislow et al, 1997); however, the underlying mechanism by which H3K4me3 maintains telomere silencing remains largely unknown.

There are two copies of four canonical histone genes in budding yeast, which are arranged in an opposite orientation to its interaction partner to coordinate their transcription (Kurat et al, 2014b). The transcription of four core histones (H2A, H2B, H3, H4) is repressed by the histone regulation (HIR) complex composed of Hir1, Hir2, Hir3, and Hpc2 to prevent the accumulation of excess histones, which could be toxic to cells (Zimmermann et al, 2018). The HIR complex interacts with Rtt106 and Asf1 to recruit the chromatin remodeling complex (RSC) to assemble repressive chromatin over histone gene promoters outside S phase (Ferreira et al, 2011). We have reported that Set1-catalyzed H3K4me3 acts as a boundary to antagonize the spread of HIR/Asf1/Rtt106 complex to promote histone gene expression during S phase (Mei et al, 2019). Histone gene transcription is activated by transcription factors or coactivators, such as Spt21, Spt10, SBF, MBF, Rtt109, and SWI/SNF (Kurat et al, 2014a; Mei et al, 2017).

The four core histone proteins are reduced and nucleosome occupancy is decreased by 50% across the genome during yeast replicative aging, which refers to the number of times a single yeast

State Key Laboratory of Biocatalysis and Enzyme Engineering, College of Life Sciences, Hubei University, Wuhan, China

Correspondence: yuxilan@hubu.edu.cn
*Qianyun Mei and Qi Yu contributed equally to this work

cell can divide (Hu et al, 2014). The replicative aging-coupled histone loss results in elevated levels of DNA strand breaks, mitochondrial DNA transfer to the nuclear genome, large-scale chromosomal alterations, translocations, and retrotransposition (Hu et al, 2014). Increased histone supply can efficiently extend the replicative life span of budding yeast (Feser et al, 2010). We have reported that reduced histones can shorten the chronological life span, which is defined as the length of time that a postmitotic cell survives, and increasing histone supply can significantly extend chronological life span (Mei et al, 2019). Moreover, a global loss of silent heterochromatin was observed during aging process in *Drosophila, Caenorhabditis elegans*, and mammalian cells (Chandra et al, 2015; Zhang et al, 2015; Wang et al, 2016). However, little is known about how chromatin structure and telomere silencing are changed during yeast chronological aging.

In this study, we reveal a novel mechanism by which four core histones regulate telomere silencing. By screening the histone mutant library, we identify eight histone mutants and Set1-catalyzed H3K4me3 that regulate telomere silencing via the core histones–autophagy–Sir2 axis. This mechanism not only exists in growing cells but also in cells that have initiated chronological aging when core histones and telomere silencing are gradually lost. Our study thus provides a mechanistic insight into regulation of telomere structure by core histones during the chronological aging process.

## Results

### Core histone levels tightly regulate telomere silencing

To address whether there is a causal relationship between intracellular core histone levels and telomere silencing, we examined telomere silencing in the H3/H4 knockdown mutant (H3/H4 KD), where two genomic copies of H3 and H4 are deleted and only one copy of histone H3 and H4 genes are expressed on a pRS-based low copy number plasmid (two to five copies per cell) (Karim et al, 2013). The expression of histones H3 and H4 was reduced in H3/H4 KD cells (Fig 1A). The transcription of telomere-proximal genes but not genes located in euchromatin regions was significantly up-regulated in H3/H4 KD mutant (Fig 1A).

We also constructed a plasmid that overexpresses histones H3 and H4 from a galactose-inducible promoter (pGAL H3/H4). Cells transformed with the pGAL H3/H4 plasmid (H3/H4 OE) had more histones H3 and H4 than cells transformed with empty vector (control) when grown in galactose-containing medium (Fig 1B). Under the same conditions, the transcription of telomere-proximal genes but not euchromatic genes was significantly reduced by overexpression of histones H3 and H4 (Fig 1B). The reduced expression of telomere-proximal genes in H3/H4 overexpression (H3/H4 OE) cells was accompanied by significantly increased histone occupancy at these genes as determined by chromatin immunoprecipitation (ChIP) (Fig 1C), indicating that increasing histone supply can efficiently enhance telomere silencing.

Histone gene transcription is induced by the transcription coactivator Spt21 (Kurat et al, 2014a). Indeed, the transcription of all four core histone genes and their protein levels were reduced in

spt21Δ mutant (Figs 1D and S1A). To examine the effect of Spt21 on telomere silencing, we deleted *SPT21* in a telomere silencing reporter strain, which has the *URA3* reporter gene inserted adjacent to the left telomere of chromosome VII (Tel VII-L) (Gottschling et al, 1990). In this reporter assay, telomere silencing is reflected by cell growth on 5-fluoroorotic acid (5-FOA)–containing medium. If telomere silencing is reduced, the telomeric *URA3* will be up-regulated and its coding product will convert 5-FOA into 5-fluorouracil (5FU), which is toxic to cell growth. The growth of spt21Δ mutant on a 5-FOA plate was retarded when compared with its WT counterpart cells (Fig 1D). By analyzing the RNA-seq data for spt21Δ mutant, it was revealed that a significant fraction of genes located within 30 kb of the telomeres was up-regulated in spt21Δ mutant, and this difference was statistically significant as determined by the $\chi^2$ test (Fig S1B). Loss of Spt21 also led to increased transcription of telomere-proximal genes, and overexpression of histones H3 and H4 partly rescued the transcription silencing defects in spt21Δ mutant (Fig 1E), suggesting that the reduced histones in spt21Δ mutant are partly responsible for compromised telomere silencing.

Histone gene transcription is negatively regulated by HIR H3/H4 histone–protein chaperone complex (Hir1, Hir2, Hir3, and Hpc2) (Osley & Lycan, 1987; Green et al, 2005). Indeed, loss of Hir1, Hir2, and Hir3 led to increased histone expression levels (Figs 1F and S1C). The transcription of telomere-proximal genes but not euchromatic genes was significantly reduced in hir2Δ and hir3Δ mutants (Fig 1F). The transcription of telomere-proximal genes (*PHO11*, *YDL241W*) was significantly reduced in hir1Δ (Fig 1F). Collectively, these results suggest that intracellular core histones tightly control telomere silencing.

### Core histone levels actively regulate the global Sir2 protein levels

To understand how core histone levels regulate telomere silencing, we examined the effect of core histones on Sir2 occupancy at telomeres using ChIP. Knockdown of histones H3 and H4 led to reduced Sir2 occupancy at telomere-proximal genes but not euchromatin genes (Fig 2A). Overexpression of histones H3 and H4 significantly increased Sir2 occupancy at telomere-proximal genes but not at euchromatin genes (Fig 2B).

Interestingly, we noticed Sir2 protein levels were significantly reduced in H3/H4 KD mutant (Fig 2C). However, the SIR complex subunit Sir3 was unaffected in H3/H4 KD mutant (Fig S2A). The reduced Sir2 was also observed in spt21Δ mutant (Fig 2D). In contrast, overexpression of histones H3 and H4 increased Sir2 protein levels (Fig 2E). Consistently, the Sir2 protein levels were significantly increased in HIR complex mutants (hir1Δ, hir2Δ, hir3Δ) (Fig 2F). These data suggest that intracellular histone levels tightly control Sir2 protein levels.

We then examined whether overexpression of Sir2 can rescue the telomere silencing defects in H3/H4 KD mutant. We transformed WT and H3/H4 KD mutant with the construct that overexpresses *SIR2* under the constitutive strong *TEF1* promoter (*pTEFpro-SIR2*). The transcription of telomere-proximal genes was significantly increased in H3/H4 KD mutant; however, overexpression of *SIR2* abrogated the de-repressed transcription of these genes in H3/H4 KD mutant (Fig 2G). Together, these results demonstrate that proper core histone levels are required to maintain Sir2 expression and telomere silencing.

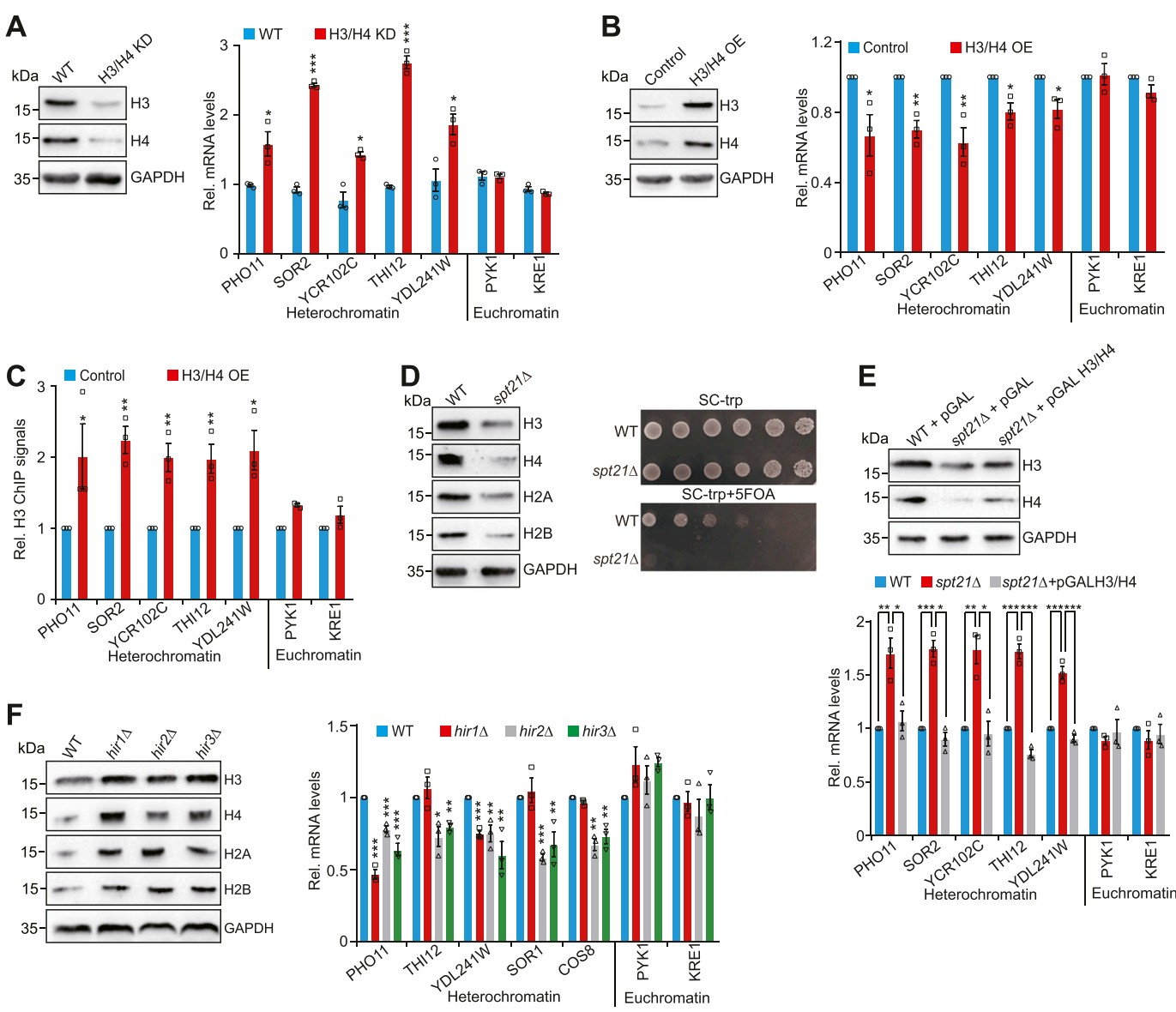

**Figure 1. Regulation of telomere silencing by core histone levels.**
**(A)** Left panel: Western blot analysis of core histones (H3, H4) in WT and H3/H4 KD cells. Right panel: qRT–PCR analysis of the transcription of *PHO11*, *SOR2*, *YCR102C*, *THI12*, *YDL241W*, *PYK1*, and *KRE1* in exponentially growing WT and H3/H4 KD cells. The mRNA levels of these genes were normalized to *ACTIN*. **(B)** Left panel: Western blot analysis of core histones (H3, H4) in WT (BY4741) cells transfected with empty vector (control) or pGAL H3/H4 (H3/H4 OE). Right panel: qRT–PCR analysis of the transcription of *PHO11*, *SOR2*, *YCR102C*, *THI12*, *YDL241W*, *PYK1*, and *KRE1* in exponentially growing control and H3/H4 OE cells. Cells were treated with 2% galactose to induce the ectopic expression of histones before being harvested for RNA extraction. **(C)** ChIP-qPCR analysis of histone H3 occupancy at telomere-proximal genes in control and H3/H4 OE cells. **(D)** WT and *spt21Δ* cells bearing *URA3* adjacent to Tel VII-L were grown to saturation, normalized for OD600, serially diluted, and spotted on SC-Trp and SC-Trp + 5-FOA plates. Impaired growth on 5-FOA plates indicates reduced silencing of *URA3*. **(E)** Top panel: Western blot analysis of core histones (H3, H4) in WT and *spt21Δ* cells transformed with empty vector (pGAL) or pGAL H3/H4. Bottom panel: qRT–PCR analysis of the transcription of *PHO11*, *SOR2*, *THI12*, *YDL241W*, *PYK1*, and *KRE1* in WT and *spt21Δ* cells transformed with empty vector (pGAL) or pGAL H3/H4. **(F)** Left panel: Western blot analysis of core histones in WT, *hir1Δ*, *hir2Δ*, and *hir3Δ* cells. Right panel: qRT–PCR analysis of the transcription of *PHO11*, *SOR2*, *YCR102C*, *THI12*, *YDL241W*, *PYK1*, and *KRE1* in exponentially growing WT, *hir1Δ*, *hir2Δ*, and *hir3Δ* cells. Data represent the mean ± SE of three independent experiments. For (A, B, C, E, F), data represent the mean ± SE of three independent experiments. *$P < 0.05$; **$P < 0.01$; ***$P < 0.001$. Source data are available for this figure.

## Reducing core histone levels accelerates autophagy-mediated Sir2 degradation

The *SIR2* mRNA levels were not significantly changed in H3/H4 KD, *spt21Δ*, and *hir1Δ* mutants (Fig S2B), suggesting that Sir2 is not regulated by core histones at the transcription level. To explore how

Sir2 is regulated in H3/H4 KD mutant, we first treated WT and H3/H4 KD cells with the proteasome inhibitor MG132 and found that the reduced Sir2 in H3/H4 KD mutant was unaffected by MG132 treatment (Figs 3A and S2C). We have previously reported that Sir2 can be degraded by autophagy (Zhang et al, 2021). We thus treated cells with PMSF, an inhibitor for lysosomal proteases. The reduced

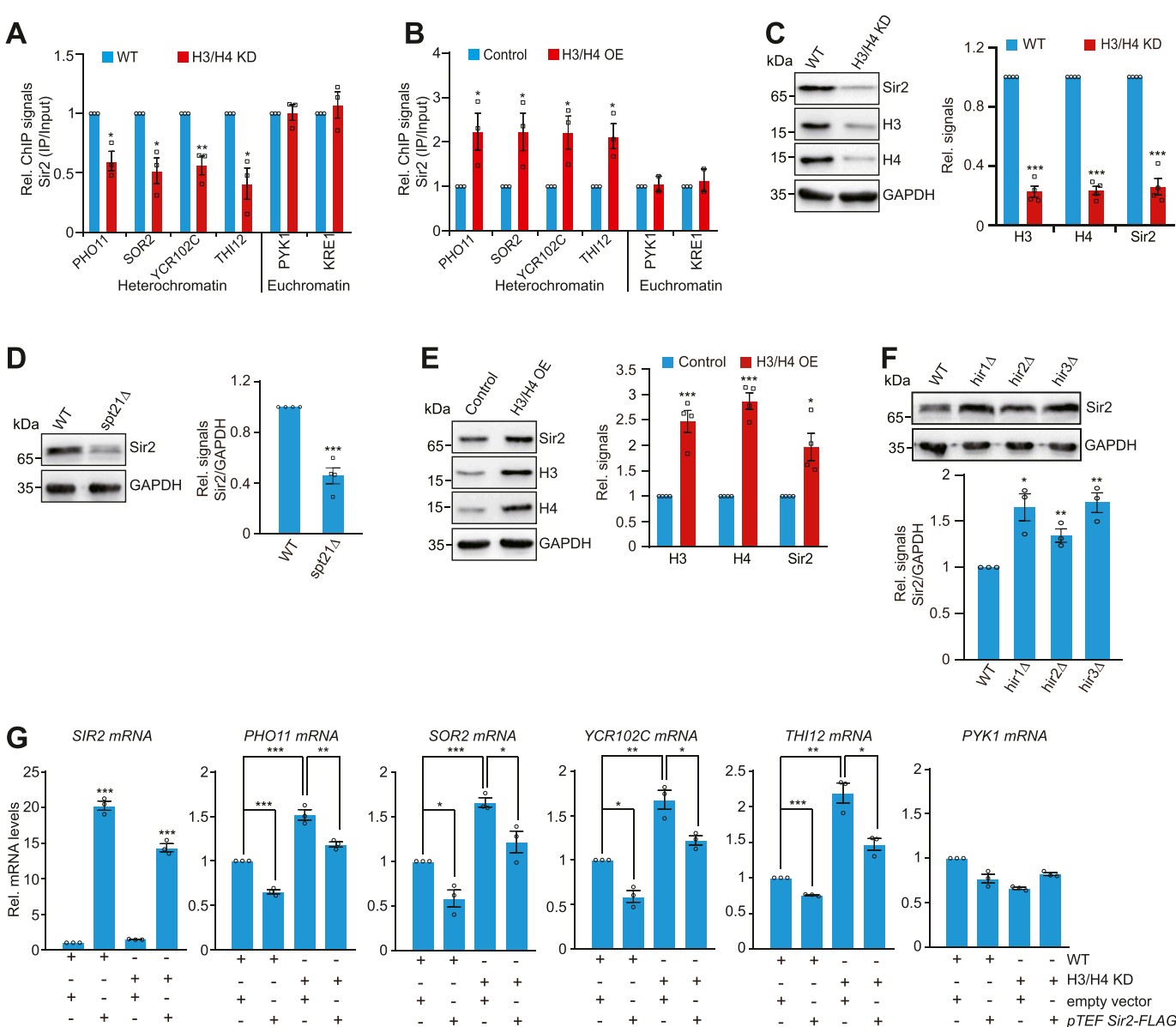

**Figure 2. Core histone levels regulate Sir2 homeostasis and Sir2 binding at telomeres.**
**(A)** ChIP-qPCR analysis of Sir2 occupancy at telomere-proximal genes in WT and H3/H4 KD cells. **(B)** ChIP-qPCR analysis of Sir2 occupancy at telomere-proximal genes in control and H3/H4 overexpressing cells (H3/H4 OE). **(C)** Western blot analysis of core histones (H3, H4) and Sir2 in WT and H3/H4 KD cells. **(D)** Western blot analysis of Sir2 in WT and *spt21Δ* mutant. **(E)** Western blot analysis of core histones (H3, H4) and Sir2 in control and H3/H4 overexpression cells (H3/H4 OE). **(F)** Western blot analysis of Sir2 in WT, *hir1Δ*, *hir2Δ*, and *hir3Δ* cells. **(G)** qRT–PCR analysis of the transcription of *SIR2*, *PHO11*, *SOR2*, *YCR102C*, *THI12*, and *PYK1* in WT and H3/H4 KD mutant transformed with empty vector (*pTEF*) or plasmid that overexpresses Sir2 (*pTEFpro-SIR2*). For (A, B, C, D, E, F, G), data represent the mean ± SE of three independent experiments. *P < 0.05; **P < 0.01; ***P < 0.001.
Source data are available for this figure.

Sir2 in H3/H4 KD mutant was restored by PMSF treatment (Figs 3A and S2D). We next treated WT and H3/H4 KD mutant with rapamycin, which is an mTOR kinase inhibitor, to induce autophagy. Rapamycin treatment reduced Sir2 in both WT and H3/H4 KD cells with H3/H4 KD mutant having less Sir2 than WT cells (Fig 3B). To further confirm that Sir2 is degraded by autophagy in H3/H4 KD mutant, we treated WT and H3/H4 KD mutant with the autophagy inhibitor chloroquine (CQ). The reduced Sir2 in the H3/H4 KD mutant was restored to normal levels by CQ treatment (Fig 3C). Moreover, deletion of the

autophagy gene *ATG2* rescued the reduced Sir2 protein in H3/H4 KD mutant (Fig 3D), suggesting that decreasing core histone levels promotes autophagy-mediated Sir2 degradation.

### Reducing core histone levels promotes the nuclear export of Sir2

Sir2 is a chromatin-binding protein located within the nucleus. We then examined Sir2 localization in WT and H3/H4 KD mutant that express a Sir2–GFP fusion protein. Fluorescence microscopy

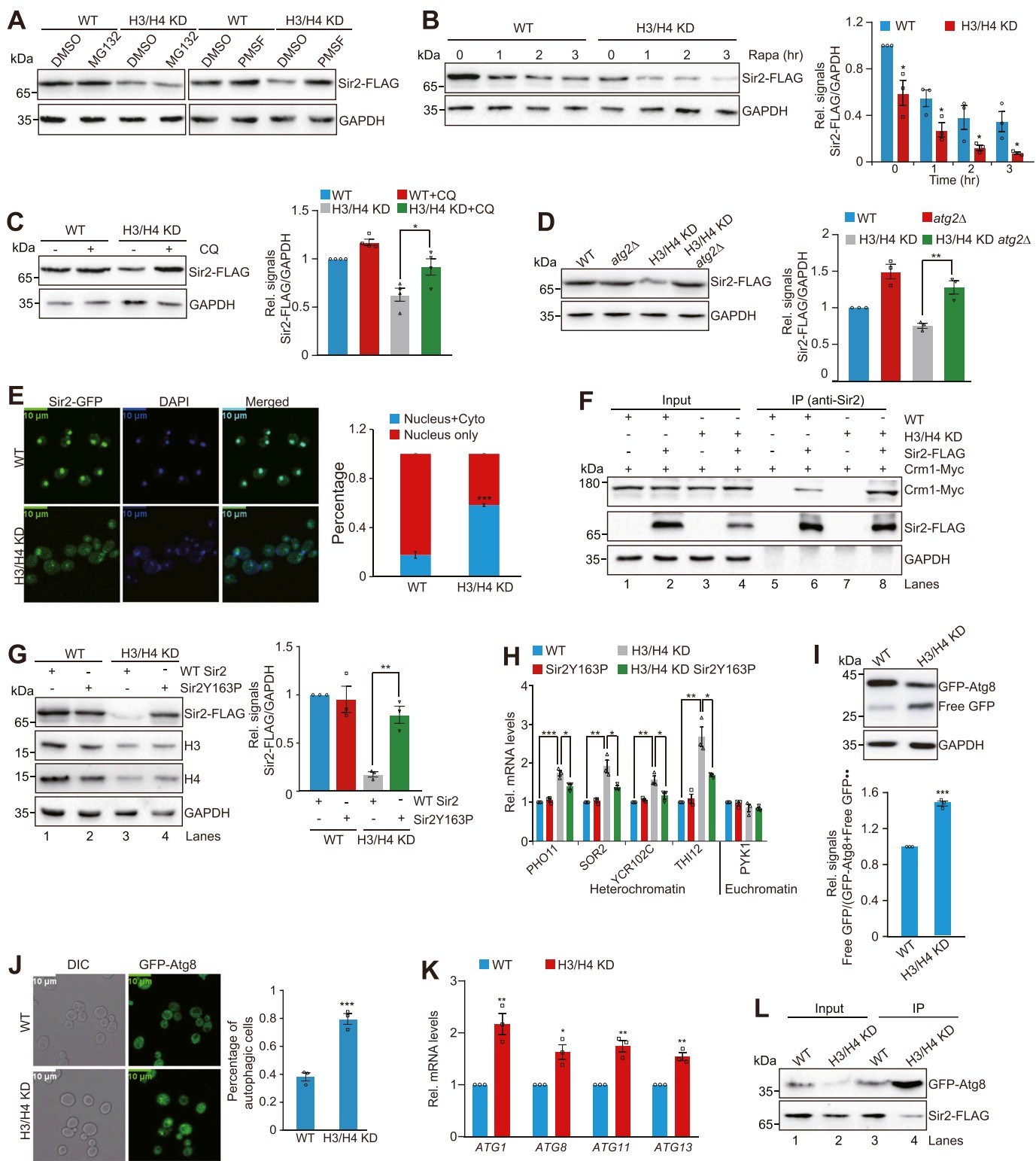

**Figure 3. Core histone levels tightly regulate autophagy-mediated Sir2 degradation.**
**(A)** Western blot analysis of Sir2 in exponentially growing WT and H3/H4 KD cells treated with 5 mM MG132 or PMSF. **(B)** Western blot analysis of Sir2 in exponentially growing WT and H3/H4 KD cells treated with 1 μg/ml rapamycin for 0, 1, 2, and 3 h. **(C)** Western blot analysis of Sir2 in exponentially growing WT and H3/H4 KD cells in the presence or absence of 10 mM CQ. **(D)** Western blot analysis of Sir2 in exponentially growing WT, atg2Δ, H3/H4 KD, and H3/H4 KD atg2Δ mutants. **(E)** Representative fluorescence images showing the distribution of Sir2–GFP (green) in WT and H3/H4 KD cells expressing Sir2–GFP from the native SIR2 locus. DAPI (blue) represented the nucleus. There was more Sir2 diffuse throughout the cell in H3/H4 KD mutant compared with WT. Bar, 10 μm. Right panel: quantification of Sir2–GFP localization in the left panel. The bar graphs represent the percentages of cells exhibiting Sir2–GFP localized in the nucleus (Nucleus only) or exported to the cytoplasm (Nucleus + Cyto). Data

https://doi.org/10.26508/lsa.202201614    vol 6 | no 3 | e202201614

analysis of Sir2–GFP in WT cells revealed that Sir2 was primarily localized in the nucleus; however, in H3/H4 KD mutant, there was a significant increase in cells that have cytoplasmic-localized Sir2 (Fig 3E), suggesting that reducing histone H3 and H4 levels promotes the nuclear export of Sir2.

Sir2 contains a consensus leucine (L)-rich nuclear export sequence L155PEDLNSLYI164, which can be recognized by the exportin chromosomal region maintenance 1 (Crm1). Using the co-immunoprecipitation assay, we detected an interaction between Sir2 and Crm1, and this interaction was enhanced in H3/H4 KD mutant (Fig 3F, lane 6 versus lane 8). Meanwhile, knockdown of CRM1 in TetO7-CRM1 mutant, where the *CRM1* promoter was replaced with TetO$_7$ and *CRM1* transcription, was shut off by doxycycline treatment (Fig S2E) (Mnaimneh et al, 2004; Yu et al, 2017), impeded rapamycin-induced degradation of Sir2 (Fig S2F). To directly show it is the dissociated Sir2 export from the nucleus that results in reduced telomere silencing in H3/H4 KD mutant, we mutated Sir2Y163P within Sir2 nuclear export sequence in WT and H3/H4 KD cells. Western blot analysis revealed that Sir2Y163P mutation rescued the reduced Sir2 in H3/H4 KD cells (Fig 3G, lane 3 versus lane 4). Strikingly, Sir2Y163P mutation partly rescued the derepressed telomere silencing in H3/H4 KD mutant (Fig 3H). All these data suggest that proper core histone levels are required to prevent Sir2 nuclear export and inhibit Sir2 degradation by autophagy, which maintains normal telomere silencing.

## Reducing core histone levels promotes the autophagic flux and enhances autophagy-mediated Sir2 degradation

We then examined the effect of intracellular core histone levels on autophagy. The GFP liberation assay demonstrated significantly increased autophagic flux with active vacuolar proteolysis in H3/H4 KD mutant compared with WT as indicated by increased ratio of free GFP/(free GFP + GFP-Atg8) in H3/H4 KD cells (Fig 3I). To strengthen these findings, we used a complementary assay by assessing the autophagy-dependent translocation of GFP-Atg8 to the vacuole by fluorescence microscopy. The percentage of autophagic cells that displayed clearly vacuolar localization of GFP was significantly increased in H3/H4 KD mutant (Fig 3J). Knockdown of H3 and H4 also significantly increased the transcription of autophagy-related genes (*ATG*), including *ATG1*, *ATG8*, *ATG11*, and *ATG13* (Fig 3K). Moreover, knockdown of histones H3 and H4 enhanced the interaction between Sir2 and Atg8 (Fig 3L, lane 3 versus lane 4).

We also examined the effect of histone regulators on autophagy. Similar to H3/H4 KD mutant, *spt21Δ* mutant had increased autophagy as determined by both GFP liberation assay and fluorescence microscopy (Fig S3A and B). In contrast, deletion of HIR complex

subunits *HIR1* and *HIR2* significantly reduced autophagy (Fig S3C and D). Consistently, analysis of RNA-seq data for *spt21Δ* mutant revealed increased *ATG* gene transcription and analysis of RNA-seq data for *hir1Δ*, *hir2Δ*, and *hir3Δ* mutants revealed reduced *ATG* gene transcription (Fig S3E–G). Together, these data indicate that intracellular core histone levels tightly control autophagy and Sir2 homeostasis.

## Screen for histone residues that regulate telomere silencing via the core histones–autophagy–Sir2 axis

Next, we aimed to identify histone residues and/or modifications that regulate telomere silencing via the core histones–autophagy–Sir2 axis. By screening the yeast H3/H4 histone mutant library, we have previously identified a total of 15 mutants with reduced histone proteins, including H3R2A, H3K4A, H3T6A, H3K14A, H3R17A, H3R40A, H3R49A, H3R53A, H3K56A, H3R69A, H3R72A, H3F104A, H4L37A, H4K44A, and H4R55A (Mei et al, 2019). To examine their effects on telomere silencing, we individually introduced these mutations into the telomere silencing reporter strain. Histones H3 and H4 were reduced in these strains (Fig 4A), consistent with our previous results (Mei et al, 2019). The telomere silencing reporter assay showed that 13 out of 15 histone mutants had impaired growth on 5-FOA plates, including H3R2A, H3K4A, H3T6A, H3K14A, H3R17A, H3R40A, H3R49A, H3K56A, H3R69A, H3R72A, H4L37A, H4K44A, and H4R55A (Fig 4B). Although histone protein levels were reduced in H3R53A and H3F104A, these two mutants had little or no effect on telomere silencing (Fig S4A). We also analyzed the RNA-seq data for some of these mutants, including H3R2A, H3R17A, H3R40A, H3R49A, H3R72A, H4K44A, and H4R55A. A significant fraction of up-regulated genes in these mutants was clustered near the telomeres (Fig S4B–H).

We then examined the effect of the above 13 histone mutants on Sir2 homeostasis. Sir2 was significantly reduced in 10 histone mutants, including H3R2A, H3K4A, H3T6A, H3K14A, H3R17A, H3R49A, H3K56A, H3R72A, H4K44A, and H4R55A (Fig 4D and E). The remaining three histone mutants H3R40A, H3R69A, and H4L37 had no effect on Sir2 protein levels despite Sir2 occupancy at telomeres being significantly reduced in these mutants (Figs 4D and E and S4I), suggesting that these three mutants reduce telomere silencing by interfering with Sir2 binding at telomere regions. Analysis of autophagy activity within the above 10 histone mutants revealed that eight histone mutants had increased autophagy, including H3R2A, H3K4A, H3T6A, H3K14A, H3R17A, H3R49A, H3K56A, and H4R55A (Fig 4F and G). To further examine whether these eight histone mutants promote autophagy-mediated Sir2 degradation, we individually treated WT and these eight mutants with autophagy

show mean ± SE from at least three experiments, with ~350 cells counted for each strain per experiment. **(F)** Co-immunoprecipitation assays showing knockdown of histones (H3/H4 KD) enhanced the interaction between the endogenously expressed Sir2 and Crm1. **(G)** Western blot analysis of Sir2 in exponentially growing WT and H3/H4 KD cells that express WT Sir2 or Sir2Y163P. **(H)** qRT–PCR analysis of the transcription of *PHO11*, *SOR2*, *YCR102C*, *THI12*, and *PYK1* in WT and H3/H4 KD mutant that express Sir2Y163P. **(I)** Western blot analysis of GFP-Atg8 and free GFP in WT and H3/H4 KD mutant expressing the endogenous *ATG8* promoter-driven GFP-Atg8 with anti-GFP antibody. **(J)** Representative fluorescence microscopy images showed the distribution of GFP-Atg8 (green) in WT and H3/H4 KD mutant. The autophagic cells were defined as cells with clear vacuolar GFP fluorescence. **(K)** qRT–PCR analysis of the transcription of *ATG1*, *ATG8*, *ATG11*, and *ATG13* in exponentially growing WT and H3/H4 KD cells. **(L)** Co-immunoprecipitation assays showing reduced core histones H3 and H4 enhanced the interaction between Sir2 and Atg8. For (B, C, D, G, H, I, J, K), data represent the mean ± SE of three independent experiments. *P < 0.05; **P < 0.01; ***P < 0.001.
Source data are available for this figure.

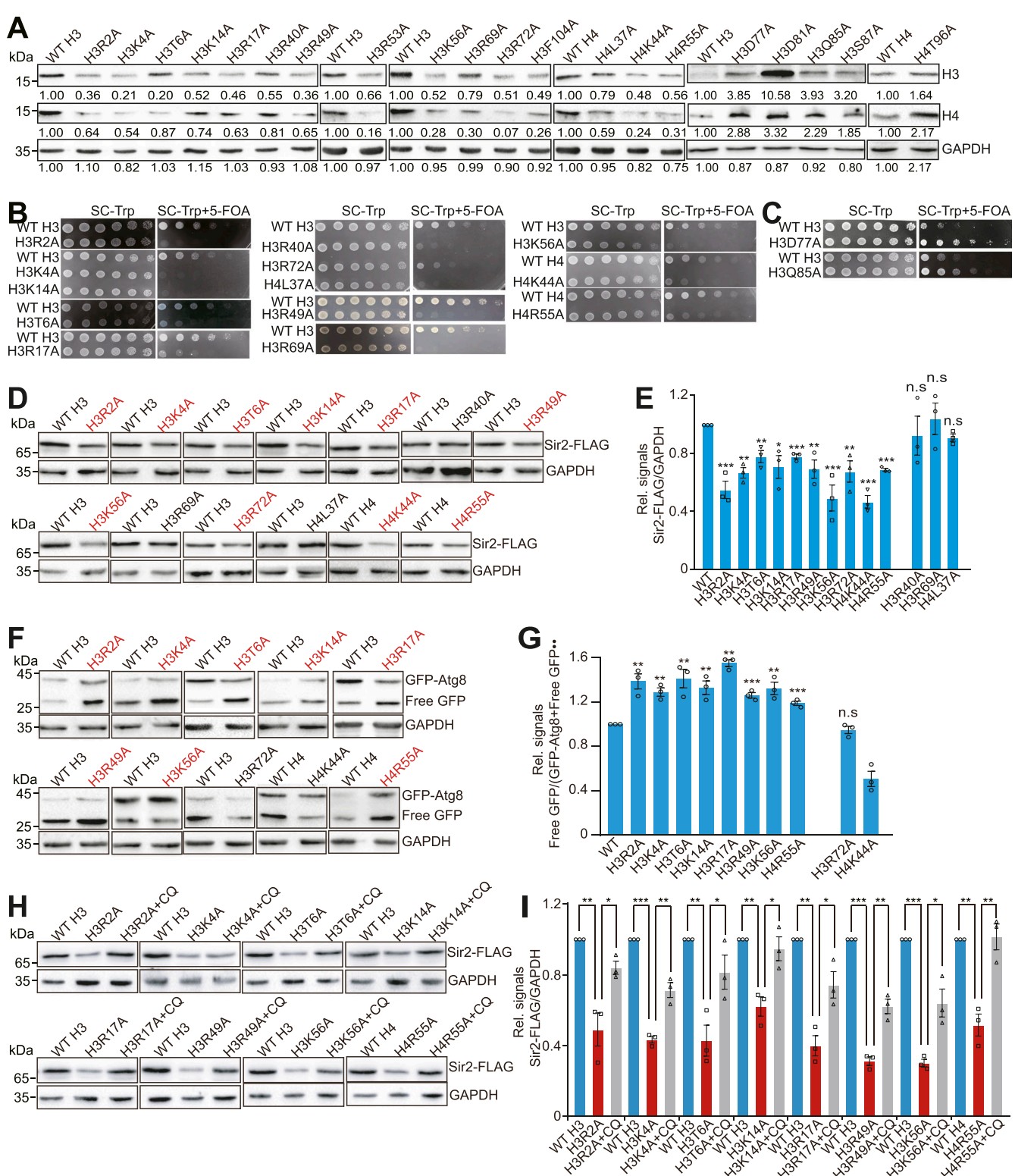

**Figure 4. Screen for histone mutants that regulate telomere silencing via the core histones–autophagy–Sir2 axis.**
**(A)** Western blot analysis of core histones in the indicated exponentially growing yeast cells. GAPDH was used as a loading control. **(B, C)** WT indicated histone mutant cells bearing *URA3* adjacent to Tel VII-L were grown to saturation, normalized for $OD_{600}$, serially diluted, and spotted on SC-Trp and SC-Trp + 5-FOA plates. The impaired growth on 5-FOA plates indicates reduced silencing of *URA3*. The better growth on 5-FOA plates indicates enhanced silencing of *URA3*. **(D, E)** Western blots and quantitation analysis of Sir2 in WT and indicated histone mutants. GAPDH was used as a loading control. **(F, G)** Western blot analysis of GFP-Atg8 and free GFP in WT and indicated histone mutants. **(H, I)** Western blot analysis of Sir2 protein in histone mutants in the presence or absence of 10 mM CQ. All experiments were performed with exponentially growing cells. For (D, E, F, G, H, I), data represent the mean ± SE of three independent experiments. *$P$ < 0.05; **$P$ < 0.01; ***$P$ < 0.001. Source data are available for this figure.

inhibitor CQ. The reduced Sir2 in these histone mutants was rescued by CQ treatment (Fig 4H and I), suggesting that these eight histone mutants promote autophagy-mediated Sir2 degradation.

There are five mutants with increased histone proteins including H3D77A, H3D81A, H3Q85A, H3S87A, and H4T96A (Fig 4A), consistent with our previous results (Mei et al, 2019). The telomere silencing reporter assay showed that only two histone mutants (H3D77A and H3Q85A) displayed better growth on 5-fluoroorotic acid plates (Figs 4C and S5A). Consistently, qRT–PCR analysis also revealed that H3D77A and H3Q85A mutants had reduced transcription of telomere-proximal genes but not euchromatin genes (Fig S5B and C). Analysis of RNA-seq data for H3D77A revealed that a significant fraction of down-regulated genes was clustered near the telomeres (Fig S5D). The occupancy of both H3 and Sir2 at telomere-proximal genes was significantly increased in these two histone mutants (Fig S5E–G). We also deleted SIR2 in these two mutants and observed that the reduced transcription of telomere-proximal genes was increased in H3D77A sir2Δ and H3Q85A sir2Δ mutants (Fig S5H and I), suggesting that H3D77A and H3Q85A mutants increase Sir2 occupancy at telomeres and enhance telomere silencing.

Altogether, we identified a total of eight histone mutants (H3R2A, H3K4A, H3T6A, H3K14A, H3R17A, H3R49A, H3K56A, H4R55A) that reduce core histone levels, promote autophagy-mediated Sir2 degradation, and compromise telomere silencing. Two histone mutants (H3R72A and H4K44A) reduce core histones, Sir2 protein levels, and telomere silencing, but they have no effect on autophagy. Three histone mutants (H3R40A, H3R69A, H4L37A) reduce Sir2 occupancy and compromise telomere silencing, but they had little effect on Sir2 protein levels. Two histone mutants (H3D77A, H3Q85A) have elevated core histones, increased Sir2 binding at telomeres, and enhanced telomere silencing (Fig S5J).

### Set1-catalyzed H3K4me3 maintains telomere silencing by repressing the core histones–autophagy–Sir2 axis

Among our screen results, we noticed H3K4A mutant with decreased core histones, increased autophagy, and reduced Sir2. We thus examined whether Set1-catalyzed H3K4me3 regulates telomere silencing via the core histones–autophagy–Sir2 axis. Mutation of H3K4A and loss of Set1 reduced the occupancy of H3K4me3 at histone genes and decreased the global levels of histone proteins (Fig S6A–C). The histone occupancy at telomere-proximal genes was also significantly reduced in H3K4A and set1Δ mutants (Fig 5A and B). Similar results were observed for Set1 complex subunit Spp1, which is specifically required for H3K4me3 (Mei et al, 2019) (Figs 5C and S6D and E).

We next examined whether Set1-catalyzed H3K4me3 affects telomere silencing. Both telomere silencing reporter assay and qRT–PCR analysis showed that the transcription of telomere-proximal genes but not euchromatic genes was significantly up-regulated in H3K4A and set1Δ mutants (Figs 5D and E and S6F and G). Analysis of RNA-seq data for set1Δ mutant also revealed that a significant fraction of up-regulated genes was clustered near the telomeres (Fig S6H), consistent with the reports that Set1 is required to maintain normal telomere silencing (Nislow et al, 1997).

To investigate whether Set1-catalyzed H3K4me3 maintains telomere silencing by promoting histone gene expression, we

overexpressed histones H3 and H4 (pGAL H3/H4) in set1Δ mutant (Fig 5F). The derepressed telomere-proximal genes in set1Δ mutant were rescued to normal levels by overexpression of core histones H3 and H4 (Fig 5G). Overexpression of core histones H3 and H4 also rescued the reduced histones and telomere silencing in spp1Δ mutant (Fig 5H and I).

We then examined the effect of Set1-catalyzed H3K4me3 on Sir2 homeostasis and autophagy. Although loss of Set1 or Spp1 had no significant effect on Sir2 mRNA level (Fig S2B), Sir2 protein level was reduced in set1Δ and spp1Δ mutants (Fig 5F and H). This reduced Sir2 can be rescued by overexpression of histones H3 and H4 (Fig 5F and H, lanes 3 and 4). Adding the autophagy inhibitor, CQ rescued Sir2 in set1Δ, H3K4A, and spp1Δ mutants (Fig 5J). We also analyzed the effect of Set1-catalyzed H3K4me3 on autophagy. The GFP liberation assay demonstrated increased autophagic flux in H3K4A, set1Δ, and spp1Δ mutants (Fig 5K and L). Collectively, these data suggest that Set1-catalyzed H3K4me3 maintains telomere silencing by repressing the core histones–autophagy–Sir2 axis.

### Core histone levels and telomere silencing are gradually lost during aging

We then examined whether the results seen in growing cells apply to cells that have initiated chronological aging. Yeast replicative aging is accompanied by reduced histone proteins (Hu et al, 2014). To understand the changes of histone protein levels during chronological aging, we grew yeast cells in YPD medium for 0, 1, 3, 6 d to initiate chronological aging. By examining the histone protein levels in cells in stationary phase, we observed a gradual loss of core histones when compared with the housekeeping genes GAPDH, actin, and tubulin, especially when cells were grown for 3 and 6 d (Fig S7A). As controls, two glycolytic enzymes Eno2 and Pgk1 were not prominently reduced during chronological aging (Fig S7A). Yeast cells in stationary phase cultures usually compose of quiescent (Q) cells and non-quiescent (NQ) cells, which can be separated by density-gradient centrifugation (Allen et al, 2006). For both NQ and Q cells, we observed a gradual loss of four core histones (Fig 6A).

This histone loss in stationary phase cells was visualized by fluorescence microscopy in the H2A-tdTomato cells, where the endogenously expressed histone H2A was tagged with tdTomato (Fig 6B–D). We also isolated the cytoplasm and chromatin fractions from cells that were grown for 0 and 3 d. By analyzing histone proteins in both fractions, we found that the chromatin-bound core histones but not histone H2A variant H2AZ were reduced in stationary phase cells (Fig 6E), indicating that the chromatinized core histones are reduced during chronological aging.

We next examined the effect of aging on telomere silencing. When cells were grown for 0, 1, 3, and 6 d, the transcription of these telomere-proximal genes was significantly increased (Fig 6F), suggesting that telomere silencing is progressively reduced when cells have initiated chronological aging. As a control, the transcription of euchromatin genes PYK1 and KRE1 was not significantly changed (Fig 6F).

As replicative aging is also accompanied by loss of histones (Hu et al, 2014), we also analyzed the effect of replicative aging on telomere silencing. By analyzing the RNA-seq data for yeast cells

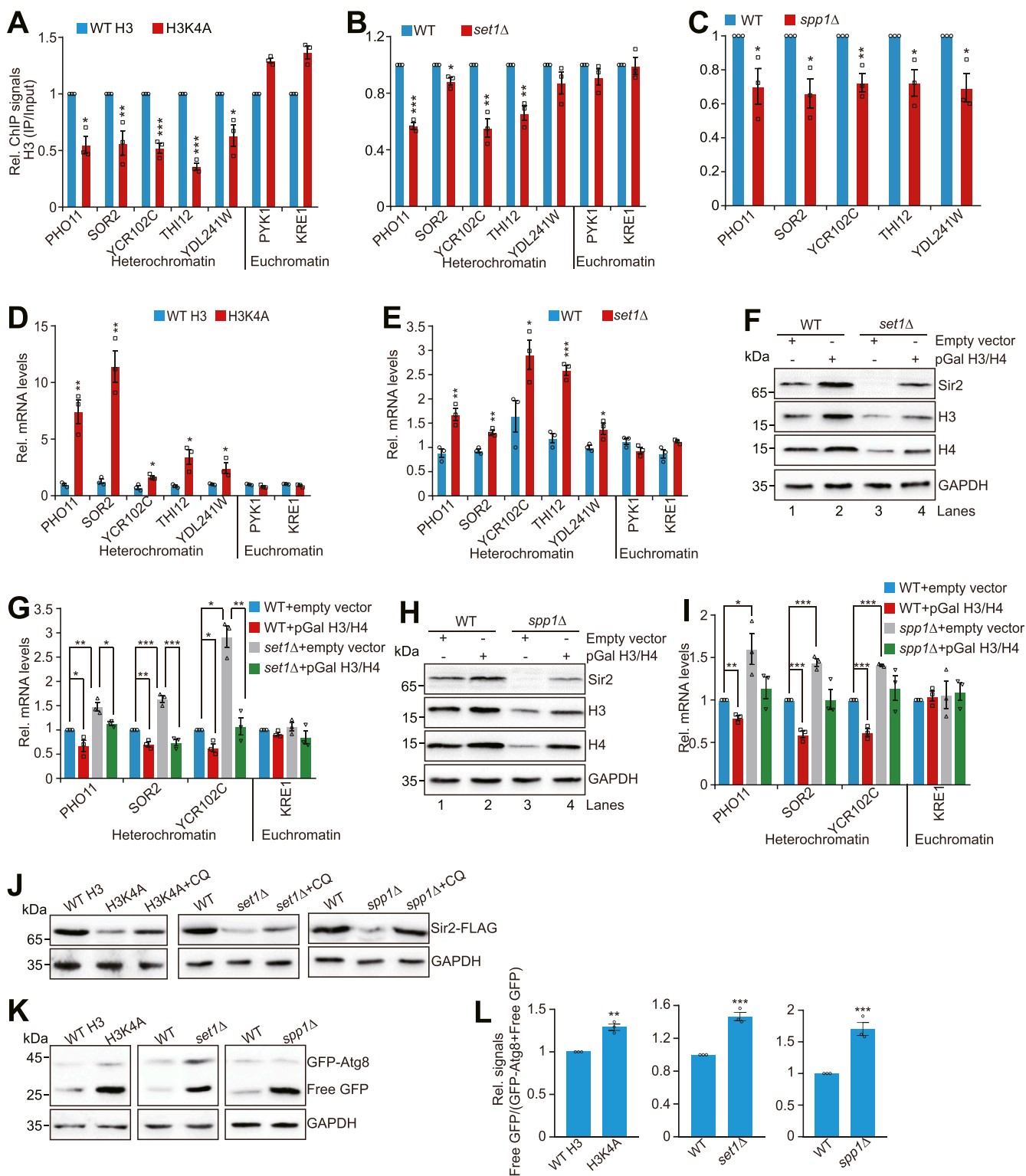

**Figure 5. Set1-catalyzed H3K4me3 maintains normal telomere silencing by repressing the core histones–autophagy–Sir2 axis.**
**(A, B, C)** ChIP-qPCR analysis of H3 occupancy in telomere-proximal genes (*PHO11, SOR2, THI12, YCR102C, YDL241W*) in exponentially growing cells (WT H3, H3K4A, WT, *set1Δ*, *spp1Δ*). *PYK1* and *KRE1* were used as negative controls. **(D, E)** The transcription of telomere-proximal genes (*PHO11, SOR2, YCR102C, THI12*, and *YDL241W*) was significantly increased in H3K4A (D) and *set1Δ* (E) mutants as determined by qRT–PCR. RNA was extracted from exponentially growing cells. The mRNA levels of all genes were normalized to *ACTIN*. The transcription of *PYK1* and *KRE1* was used as negative controls. **(F)** Western blot analysis of Sir2 and core histone proteins (H3, H4) in WT (empty vector or pGAL H3/H4), *set1Δ* (empty vector or pGAL H3/H4) cells. **(G)** qRT–PCR analysis of the transcription of telomere-proximal genes in WT (empty vector or pGAL H3/H4), *set1Δ* (empty vector or pGAL H3/H4) cells. **(H)** Western blot analysis of Sir2 and core histone proteins (H3, H4) in WT (empty vector or pGAL H3/H4), *spp1Δ* (empty vector

subjected to replicative aging for 7.5, 24, and 48 h (Cruz et al, 2018), we found a significant fraction of the genes located within 40 kb of the telomeres was induced in aged cells relative to log phased cells (Fig S7B). Moreover, the transcription of telomere-proximal genes, including *SEO1*, *SOR1*, *SOR2*, *YCR102C*, and *THI12*, which locate 7.24, 8.63, 8.68, 11.15, and 14.83 kb from the nearest telomeres, respectively, progressively increased during aging process (Fig S7C). These results indicate that both core histones and telomere silencing are gradually lost during replicative and chronological aging.

### Core histone levels tightly regulate Sir2 homeostasis and telomere silencing during chronological aging

Loss of histones accelerates chronological aging, and increasing histone supply prevents premature chronological aging (Mei et al, 2019). Loss of H3K4me3 in *set1Δ* or *spp1Δ* mutants accelerated chronological aging (Walter et al, 2014). Histone mutants like H3R2A, H3K4A, and H3T6A had shortened life span, whereas H3D77A had prolonged life span (Mei et al, 2019). In addition to these histone mutants, H3R17A had reduced histones and shortened life span (Fig S8A), whereas H3Q85A and H4T96A had increased histones and prolonged life span (Fig S8B and C). We thus determined whether core histone levels regulate Sir2 homeostasis and telomere silencing when cells have initiated chronological aging. The global levels of Sir2 were not only reduced during aging but also lower in H3/H4 KD mutant compared with WT (Fig 7A). Loss of Atg12 attenuated the reduced Sir2 in H3/H4 KD cells when chronological aging was initiated (Fig 7B), suggesting that loss of core histones accelerates autophagy-mediated Sir2 degradation during chronological aging. With reduced Sir2 protein levels, the transcription of telomere proximity genes was significantly increased when cells have initiated chronological aging (Fig 7C). Moreover, the transcription of telomere-proximal genes was significantly higher in H3/H4 KD mutant than WT (Fig 7C), which is consistent with the global changes of Sir2 (Fig 7A).

When *set1Δ* mutant was examined, we found that the overall levels of histones and Sir2 were lower than WT when cells have initiated chronological aging (Fig 7D). Deletion of *ATG12* attenuated the loss of Sir2 in *set1Δ* mutant when chronological aging was initiated (Fig S8D), suggesting that Set1 prevents autophagy-mediated degradation of Sir2 during chronological aging. In agreement with histones and Sir2 changes, the transcription of telomere-proximal genes was significantly higher in *set1Δ* mutant than WT when chronological aging was initiated (Fig 7E). Similar results were observed in H3R17A mutant (Fig 7F and G).

Taken together, these data suggest that during chronological aging, the chromatin-bound core histones are reduced, which accelerates Sir2 nuclear export and degradation by autophagy. Moreover, loss of core histones increases autophagy, which further enhances Sir2 degradation and leads to compromised telomere silencing.

## Discussion

Telomere dysfunction is an important hallmark of aging and aging-related diseases (Chakravarti et al, 2021). Maintaining the telomere structure and transcription silencing is important to prevent premature aging (Smeal et al, 1996; Kozak et al, 2010). How transcriptional silencing is dynamically regulated at telomeres during aging remains largely unknown. Here, we find that the yeast chronological aging is accompanied by a global loss of core histone proteins. Moreover, we identify the core histones–autophagy–Sir2 axis to explain how core histone levels regulate telomere silencing: core histone levels inhibit autophagy, which protects Sir2 from being degraded by autophagy and maintains normal telomere silencing. When cells have initiated chronological aging, core histone levels are gradually lost, which leads to dissociation of Sir2 from telomeres and promotes its nuclear export to the cytoplasm to be degraded by autophagy. By screening the histone mutant library, we identify eight histone mutants and one histone modification (Set1-catalyzed H3K4me3) that regulate telomere silencing via the core histones–autophagy–Sir2 axis during chronological aging. Together, our study provides a novel insight into regulation of telomere structure and transcriptional silencing by core histones during chronological aging.

A global loss of silent heterochromatin has been observed during aging process in *Drosophila*, *C. elegans*, and mammalian cells (Chandra et al, 2015; Zhang et al, 2015; Wang et al, 2016). In budding yeast, nucleosome occupancy is decreased by 50% across the genome during replicative aging (Hu et al, 2014). Here, we show that core histone levels are lost during chronological aging. Most importantly, we observed that heterochromatin is sensitive to core histone level changes (Fig 1C). Although it remains unclear about the precise mechanism, the existence of multiple histone chaperones, including Spt6, FACT, Rtt106, and Asf1 may ensure the recycling of histones at actively transcribed regions (Smolle & Workman, 2013). Moreover, some chromatin remodelers, such as Chd1 and Isw1b, may also maintain nucleosome occupancy in transcribed regions (Smolle & Workman, 2013). The specific loss of core histones at heterochromatin reduces the binding of Sir2 and probably SIR complex at heterochromatin, which may in turn accelerate core histone loss from heterochromatin.

Our data reveal an active function of core histone levels in regulating telomere silencing by controlling autophagy and Sir2 homeostasis. During both replicative and chronological aging, telomere silencing is compromised and genes located within subtelomere regions are derepressed. By combining telomere silencing reporter assays, qRT–PCR and RNA-seq, we show that core

---

or pGAL H3/H4) cells. **(I)** qRT–PCR analysis of the transcription of telomere-proximal genes in WT (empty vector or pGAL H3/H4), *spp1Δ* (empty vector or pGAL H3/H4) cells. **(J)** Western blot analysis of Sir2 in WT H3, H3K4A, *set1Δ*, and *spp1Δ* mutants treated with or without 10 mM CQ. **(K, L)** Western blot analysis of GFP-Atg8 and free GFP in WT H3, H3K4A, WT, *set1Δ*, and *spp1Δ* cells. Data represent the mean ± SE of three independent experiments. For (A, B, C, D, E, G, I, K, L), data represent the mean ± SE of three independent experiments. *P < 0.05; **P < 0.01; ***P < 0.001.
Source data are available for this figure.

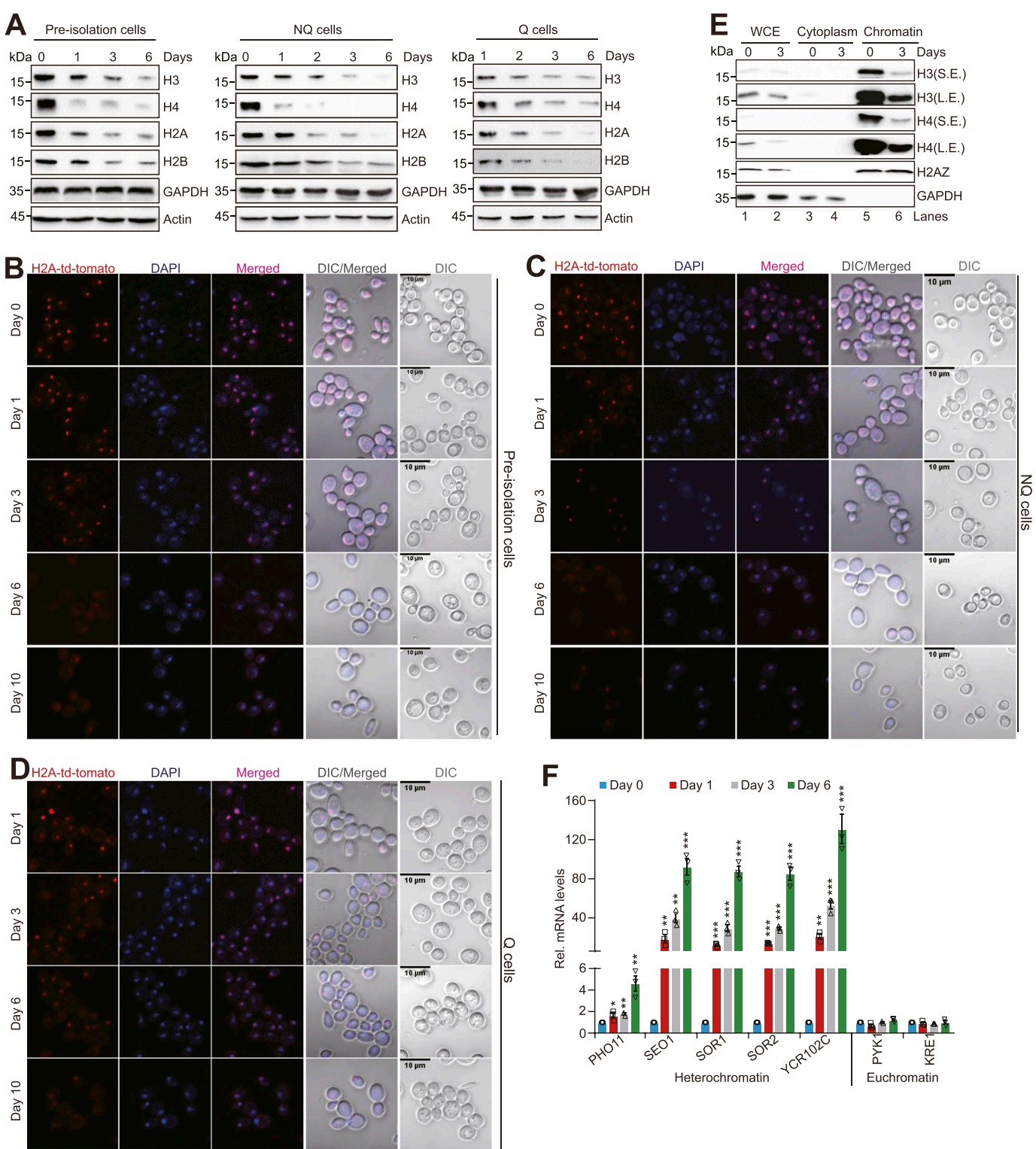

**Figure 6.  Core histones are reduced during chronological aging.**
**(A)** Western blot analysis of core histones in pre-isolation cells (heterogenous stationary phase cells), NQ cells, and Q cells when grown for 0–6 d in YPD medium. NQ and Q cells were separated by the Percoll density-gradient centrifugation. **(B, C, D)** Representative fluorescence microscopy images show the loss of histones in NQ cells, Q cells, and pre-isolation cells when grown for 0–6 d in YPD medium. Red color indicates the endogenously expressed histone H2A tagged with tdTomato. Blue color indicates DAPI staining. Scale, 10 μm. **(E)** Effects of chronological aging on chromatin-bound histones by Western blots. Total (WCE) cytoplasm and chromatin-bound proteins were extracted from cells when grown for 0 and 3 d. S.E., short exposure; L.E., long exposure. **(F)** qRT–PCR analysis of the transcription of *PHO11*, *SEO1*, *SOR1*, *SOR2*, *YCR102C*, *PYK1*, and *KRE1* in WT (BY4741) cells when grown for 0, 1, 3, and 6 d in YPD medium. The mRNA levels of these genes were normalized to *ACTIN*. Data represent the mean ± SE of three independent experiments. *$P < 0.05$; **$P < 0.01$; ***$P < 0.001$.
Source data are available for this figure.

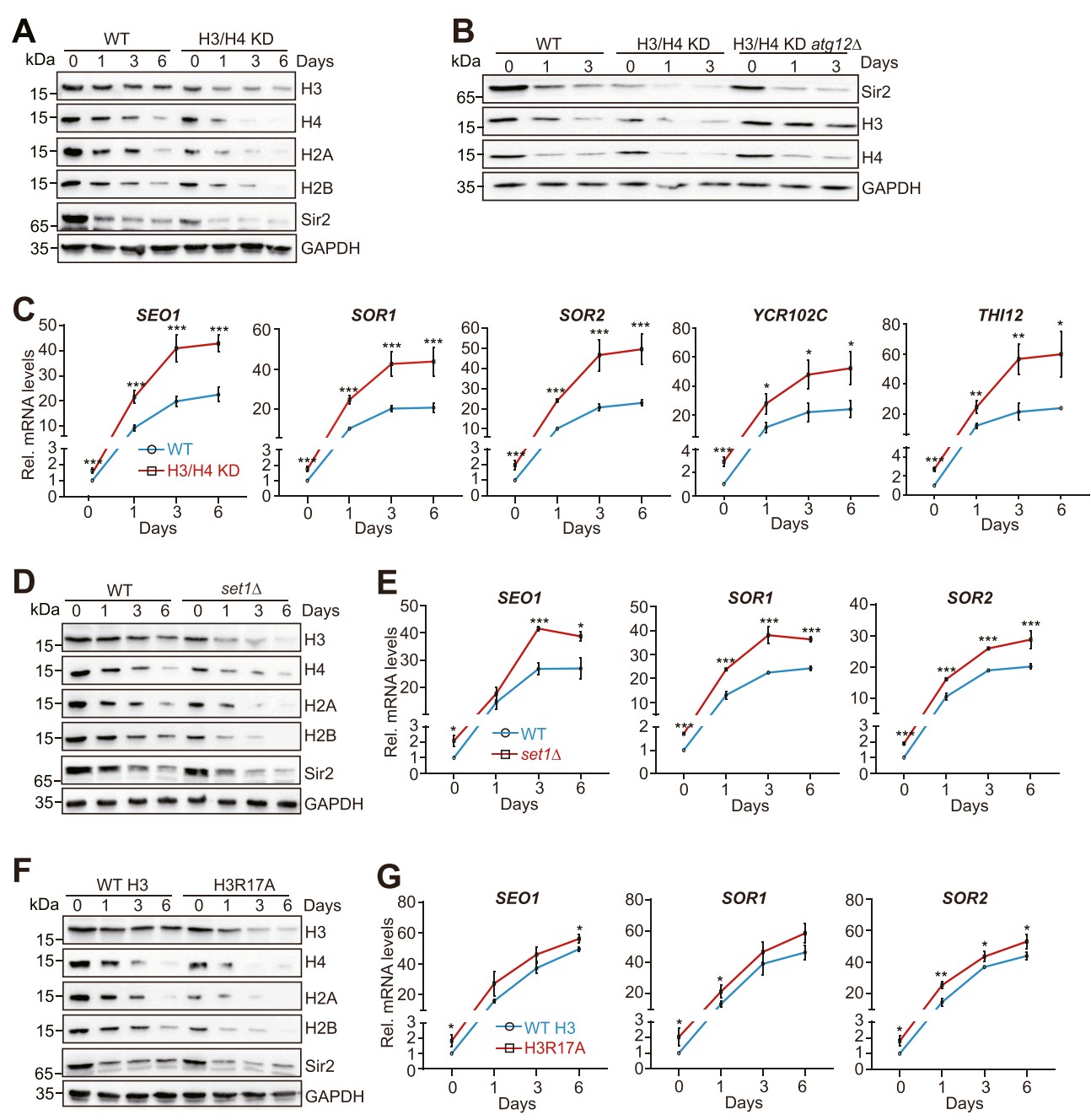

**Figure 7. Regulation of telomere silencing by the core histones–autophagy–Sir2 axis during chronological aging.**
**(A)** Western blot analysis of core histones and Sir2 in WT and H3/H4 KD cells when grown for 0–6 d in YPD medium. **(B)** Western blot analysis of core histones (H3, H4) and Sir2 in WT, H3/H4 KD, and H3/H4 KD *atg12Δ* cells when grown for 0–3 d in YPD medium. **(C)** qRT–PCR analysis of the transcription of *SEO1*, *SOR1*, *SOR2*, *YCR102C*, and *THI12* in WT and H3/H4 KD cells when grown for 0–6 d in YPD medium. The RNA levels of these genes were normalized to *ACTIN*. **(D, E)** Analysis of core histones, Sir2, and telomere silencing in WT and *set1Δ* cells when grown for 0–6 d in YPD medium. **(F, G)** Analysis of core histones, Sir2, and telomere silencing in WT and H3R17A cells when grown for 0–6 d in YPD medium. For (C, E, G), data represent the mean ± SE of three independent experiments. *P < 0.05; **P < 0.01; ***P < 0.001.
Source data are available for this figure.

histone levels play an active role in regulating telomere silencing. Furthermore, analysis showed that core histone levels tightly regulate Sir2 homeostasis. Loss of core histones leads to dissociation of Sir2 from telomeres and nuclear export of Sir2, which results in autophagy-mediated Sir2 degradation and reduced telomere silencing. Moreover, knockdown of histones enhances autophagy, which further accelerates Sir2 degradation by autophagy. During chronological aging, the reduced core histones contribute to compromised telomere silencing partly, if not completely, by promoting autophagy-mediated Sir2 degradation. As core histones and telomere silencing are progressively reduced during replicative aging, it is highly likely that core histones might

regulate telomere silencing by the same pathway during replicative aging. Although it is possible that telomere silencing defects could be caused directly by a loss of core histones and reduced nucleosome intensity, we indeed observe that telomere silencing was restored in the Sir2Y163P mutant that blocks nuclear export, suggesting that core histones maintain telomere silencing partly, if not completely, by controlling Sir2 homeostasis.

Recently, the Foiani group reported that upon glucose starvation conditions, Rad53 reduces the transcription of histones to activate the transcription of stress response genes located within the subtelomere regions (Bruhn et al, 2020), which is consistent with our observation that reducing core histones compromised telomere silencing. In the absence of Rad53, the transcription of Sir3 is increased and deletion of *HHT2*, which encodes histone H3, reduces Sir3 to normal levels (Bruhn et al, 2020), suggesting that regulation of Sir3 expression is specific for Rad53 under specific conditions. Here, we find that reducing core histone levels via knockdown of histones H3 and H4 has no effect on Sir3 expression but significantly reduces Sir2 protein level via autophagy. Analysis of *spt21Δ* RNA-seq data also reveals that the transcription of Sir2 (1.08-fold change) and Sir3 (1.05-fold change) is unaffected by Spt21 depletion. These data suggest that under normal conditions, core histones regulate telomere silencing primarily, if not all, by controlling autophagy-mediated Sir2 degradation but not Sir3 expression. Therefore, we uncover a novel pathway for regulation of telomere silencing by intracellular core histone levels. As SIRT1, the Sir2 homolog in mammals, has recently been reported to undergo autophagy during aging (Xu et al, 2020), it is possible that core histones also regulate SIRT1 homeostasis in mammals.

By screening the histone substitution mutant library, we identify a total of 13 histone mutants having reduced histones and compromised telomere silencing. These mutants include H3R2A, H3K4A, H3T6A, H3K14A, H3R17A, H3R40A, H3R49A, H3K56A, H3R69A, H3R72A, H4L37A, H4K44A, and H4R55A. We also identified two histone mutants (H3D77A and H3Q85A) that have increased core histones and enhanced telomere silencing. These 15 histone residues localize at three primary structure clusters: five histone residues (H3R2, H3K4, H3T6, H3K14, H3R17) localize on the H3 N-terminal tail; the second cluster is composed of H3R40, H4K44, H3R49, H3K56, and H4L37; the third cluster consists of H4R55, H3R69, H3R72, H3D77, and H3Q85 (Fig S5J). Among these mutations, four histone mutants (H3R2A, H3K4A, H3K14A, and H3R72G) have been shown to regulate telomere silencing (Park et al, 2002; Thompson et al, 2003; Kirmizis et al, 2007; Prescott et al, 2011). H3D77N mutation has been reported to suppress the telomere silencing defects caused by Sir3 BAH domain (Sampath et al, 2009). As H3D77A caused increased histones at subtelomere regions and represses the transcription of telomere-proximal genes, our study provides one possible mechanism for how H3D77N represses telomere silencing defects. Among these 13 histone mutants with reduced histones and compromised telomere silencing, we identify a total of 8 (61.5%) histone mutants (H3R2A, H3K4A, H3T6A, H3K14A, H3R17A, H3R49A, H3K56A, H4R55A) that reduce telomere silencing by promoting the core histones–autophagy–Sir2 axis. Two histone mutants (H3R72A and H4K44A) reduce Sir2 in a manner independent of autophagy, which may regulate Sir2 expression at the transcriptional level. Three histone

mutants (H3R40A, H3R69A, H4L37) reduce Sir2 occupancy and compromise telomere silencing but have little effect on Sir2 protein levels. In addition, some histone mutants, that is, H3R17A, have reduced core histones, telomere silencing during chronological aging, which may explain how they shorten chronological life span. Thus, our study provides a novel mechanistic insight into regulation of telomere silencing and probably aging by histone mutants.

Set1-catalyzed H3K4me3 has been reported to be required to maintain telomere silencing (Nislow et al, 1997; Corda et al, 1999; Fingerman et al, 2005). Loss of H3K4me3 leads to reduced chronological life span (Mei et al, 2019; Yu et al, 2022). In this study, we show that Set1-catalyzed H3K4me3 maintains telomere silencing via the core histones–autophagy–Sir2 axis. Loss of Set1 and H3K4me3 results in reduced intracellular core histones and increased autophagy-mediated Sir2 degradation. This mechanism also applies to the aging process. Moreover, histone posttranslational modifications that specifically affect H3K4me3 may regulate this pathway. These modifications include H3R2 asymmetric dimethylation (H3R2me2a), H3T6 phosphorylation (H3T6ph), and H3K14 acetylation (H3K14ac) as these modifications have been reported to regulate H3K4me3 (Kirmizis et al, 2007; Metzger et al, 2010; Maltby et al, 2012). H3R2A, H3T6A, and H3K14A mutants have reduced histones and compromised telomere silencing. In addition, H3R2A, H3K4A, H3T6A, and H3K14A mutants have accelerated autophagy and reduced Sir2 degradation. It is highly likely that H3R2me2a, H3T6ph, and H3K14ac could regulate telomere silencing by modulating the core histones–autophagy–Sir2 axis. Hence, we unveiled an epigenetic mechanism for how Set1-catalyzed H3K4me3 and its regulatory histone PTMs regulate telomere silencing.

Yeast telomeres contain Y′ repeats and variable X sequences, the numbers of which may affect the distance between the genes being assayed and the nearest telomere and eventually influence their silencing. We thus performed Southern blots to detect Y′ and terminal telomere fragments in the mutants examined in this study (Shampay & Blackburn, 1988; Tsukamoto et al, 2005). Cells were collected and DNA was cut with *Xho*I and probed with a Y′-TG probe that gave three fragments, 6.5, 5.5, and 1.3 kb in WT cells. The two larger fragments correspond to repetitive Y′, whereas the shortest fragment is the terminal 1 kb of Y′ and also contains ~350 bp of telomeric TG repeats. Our data showed that overexpression or knockdown of histones H3/H4 had no effect on telomeric fragment (Fig S8E). In addition, the telomere length in *spt21Δ*, *hir1Δ*, *hir2Δ*, *hir3Δ*, *set1Δ*, and *spp1Δ* was indistinguishable from WT (Fig S8E). For histone mutations, H3R2A, H3K4A, H3T6A, H3K14A, H3R40A, H3R49A, H3K56A, H3R72A, H3D77A, H3Q85A, H4L37A, H4K44A, and H4R55A had similar telomere length with their WT counterpart (Fig S8F). These data suggest that the effect of these protein or histone mutants on telomere silencing is not because of major changes in length of the terminal telomere repeat tract or the number of Y′ repeats.

Collectively, we presented here a detailed analysis of telomere structure regulation during aging by core histones and autophagy. Based on our findings, blocking the core histones–autophagy–Sir2 axis can help attenuate the aging-coupled loss of telomere silencing and may prevent premature aging, which may provide clues

into development of anti-aging strategy and reducing the incidence of aging-related diseases.

# Materials and Methods

### Yeast strains

All yeast strains used in this study are described in Table S1. The histone mutants were constructed by transforming the plasmids that express histone H3/H4 mutants into the telomere silencing reporter strain UCC1369 and then selecting on α-aminoadipic acid media for cells that had lost the WT H3 and H4 as described (Mei et al, 2019).

### Quantitative reverse transcription PCR (qRT–PCR)

RNA was isolated from each sample by standard phenol–chloroform extraction procedures. The quality of RNA was analyzed by agarose gel electrophoresis and quantified by a NanoDrop spectrophotometer. Purified RNA was then subjected to quantitative reverse transcription PCR (qRT–PCR) as described previously (Yu et al, 2017). Results were analyzed using $2^{(-\Delta\Delta Ct)}$. Primers used for qRT-PCR are listed in Table S2.

### Preparation of yeast whole cell extracts and Western blot analysis

Proteins were extracted from yeast cells as previously described (Li et al, 2015). Cells grown in 3–5 ml culture were harvested and lysed in 2 M NaOH with 8% 2-mercaptoethanol. Compared with the exponentially growing cells, the aged cells were lysed for longer time. Cell lysate was centrifuged and the pellet was washed with the washing buffer (40 mM HEPES-KOH, pH 7.5; 10% glycerol; 350 mM NaCl; 0.1% Tween 20). Cell pellets were resuspended in 2×SDS sample buffer and boiled at 95°C for 10 min. After centrifugation, the cleared protein samples were separated by 15% SDS–PAGE and transferred to Immobilon PVDF membrane (Bio-Rad). The blots were probed with antibodies against specific proteins followed by incubation with horseradish peroxidase–labeled IgG secondary antibodies or IRDye 680RD goat anti-mouse and IRDye 800CW goat anti-rabbit secondary antibodies. The specific proteins were visualized by using the ECL Chemiluminescence Detection Kit (170-5061; Bio-Rad) or scanned with a Li-Cor Odyssey infrared imaging system (Mei et al, 2019). All antibodies used are listed in Table S3. The specificity of customized anti-Sir2 antibody was verified by Western blots on cell lysates of WT and sir2Δ mutant (Fig S2A).

### Cell fractionation

Percoll density gradients (GE Healthcare) were prepared using the manufacturer's protocol with modifications. Percoll was diluted 9:1 (vol/vol) with 1.5 M NaCl. 1.8 ml of the Percoll solution was put into a 2-ml tube and centrifuged at 17,885$g$ for 15 min at 20°C to form the gradients. $1 \times 10^9$ cells were pelleted, resuspended in 0.2 ml 0.5 M Tris buffer, overlaid onto the preformed gradient, and centrifuged

at 400$g$ for 1 h at 20°C. The upper fraction cells are non-quiescent (NQ) cells, and the lower fraction cells are quiescent (Q) cells. Fractions were collected, washed once in 1 ml 0.5 M Tris buffer, pelleted, and resuspended in ddH$_2$O for subsequent assays.

### ChIP assay

The ChIP assays were performed as previously described (Li et al, 2015; Wu et al, 2019). Yeast cells were grown in 200 ml YPD media at 28°C until OD$_{600}$ of 0.7–1.0. Crosslinking was performed by adding 5.6 ml 37% formaldehyde for 15 min and quenched by adding 10 ml of 2.5 M glycine. Cells were then harvested, washed, and lysed with FA-SDS lysis buffer (0.1% SDS; 40 mM HEPES-KOH, pH 7.5; 1 mM EDTA, pH 8.0; 1% Triton X-100; 0.1% Na deoxycholate; 1 mM PMSF; 2 μg/ml leupeptin; 1 μg/ml pepstatin A; Sigma-Aldrich protease inhibitor cocktail). DNA was sonicated into ~500 bp fragments and immunoprecipitated with anti-H3 (2 μl; ab1791; Abcam), which was pre-bound to Dynabeads. Beads were washed sequentially with FA lysis buffer, FA buffer with 500 mM NaCl, TEL buffer (10 mM Tris, pH 8.0; 1 mM EDTA; 0.25 M LiCl; 1% NP-40; 1% Na deoxycholate), and TE (10 mM Tris, pH 7.4; 1 mM EDTA). The DNA/chromatin complexes were eluted twice with the elution solution (10 mM Tris, pH 8.0; 1 mM EDTA; 1% SDS; 250 mM NaCl) at 65°C for 30 min followed by treatment with 20 μg Proteinase K (Roche) at 55°C for 1 h and reverse crosslinking at 65°C overnight. The DNA was digested with RNase A, precipitated with ethanol, and quantitated by qRT-PCR. Primers used for qRT-PCR are listed in Table S2.

### Chronological aging experiments

The yeast chronological life span was performed as described previously (Mei et al, 2019; Chen et al, 2021). Briefly, a total of 5 ml of seed cultures from individual colonies were grown overnight at 30°C in YPD medium. Cells were then cultured into 50 ml media with an initial OD$_{600}$ of 0.1 and grown at 30°C for 3 d until cell growth ceased. On day 3, 100 μl aliquots from each culture were collected; 10-fold serial dilution was created and spread on YPD plate to determine the CFU. On the following days, 100 μl aliquots were regularly removed from the remaining cultures, diluted, plated, and counted. The life span curve was created by plotting the CFU score for each culture as a function of time.

### Telomere silencing reporter assays

Telomere silencing reporter assays were performed as described previously (Zhang et al, 2021). Strain UCC1365 and derived mutants were grown overnight to saturation in YPD, normalized for OD$_{600}$, serially diluted, and spotted on plates with or without 5-fluoroorotic acid (100 μg/ml).

### Analysis of telomere length and Southern blot

The genomic DNA was extracted from 100 OD$_{600}$ yeast cells. RNA was removed by digestion with RNase A. Genomic DNA was digested with XhoI overnight and electrophorized on a 1% agarose gel. DNA was then transferred to Hybond-N membrane and UV crosslinked.

The membrane was hybridized with digoxigenin-labeled Y′-TG probe (5′-CAGTTTAGCAGGCATCATCG-3′; 5′-CGAGAACTTCAACGTTTGCC-3′) and detected using chemiluminescence.

## Quantification and statistical analysis

The Image J software was used to quantify the Western blots' data by measuring the relative intensity of each band. Quantification data were presented as the mean ± SE from at least three independent experiments. Statistical differences in this study were determined by two-tailed unpaired $t$ test, and a $P$-value < 0.05 was considered statistically significant.

For analysis of telomere-proximal genes, genes within 60 kb of telomeres were ordered according to their distance from telomeres (Chen et al, 2021). The genes were grouped by their distance from the nearest telomeres in consecutive intervals of 10 kb. The $\chi^2$ values score the significance of the difference between the expected frequency from the genome-wide average to the observed frequency in each 10 kb interval with the degree of freedom of one. For each interval, the fraction of genes that were differentially expressed in each mutant was calculated and listed in Table S4. The statistical significance is defined as the following: $\chi^2 > 3.841$, $P < 0.05$; $\chi^2 > 6.635$, $P < 0.01$; $\chi^2 > 10.828$, $P < 0.001$.

# Data Availability

The accession numbers for the RNA-seq dataset used in this article are GSE107744 (young and aged cells), GSE110003 (*spt21Δ*), GSE42526 (*hir1Δ*, *hir2Δ*, *hir3Δ*), GSE73407 (*set1Δ*), GSE29059 (H3R2A, H3R17A, H3R40A, H3R49A, H3R72A, H4K44A, H4R55A, H3D77A, H3D81A). The differential expression levels of aligned sequences were calculated, and genes with a fold change > 1.5 and adjusted $P < 0.05$ were used for further analysis.

# Supplementary Information

# Acknowledgements

We sincerely thank the lab members for discussion of this project and critical reading of the article. This project was funded by grants from National Natural Science Foundation of China (31872812) to X Yu and Natural Science Foundation of Hubei Province (2019CFA077) to X Yu.

## Author Contributions

Q Mei: data curation, formal analysis, validation, investigation, methodology, and writing—original draft, review, and editing.
Q Yu: data curation, software, formal analysis, validation, investigation, methodology, and writing—original draft, review, and editing.
X Li: investigation and methodology.
J Chen: formal analysis and methodology.
X Yu: conceptualization, data curation, formal analysis, supervision, funding acquisition, investigation, visualization, project administration, and writing—original draft, review, and editing.

## Conflict of Interest Statement

The authors declare that they have no conflict of interest.

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
