## [Reviewer comments · Life Science Alliance]

Life Science Alliance

Regulation of telomere silencing by the core histones-autophagy-Sir2 axis

Qianyun Mei, Qi Yu, Xin Li, Jianguo Chen, and Xilan Yu

DOI: <https://doi.org/10.26508/lsa.202201614>

Corresponding author(s): Xilan Yu, Hubei University

Review Timeline:

Submission Date:	2022-07-19
Editorial Decision:	2022-09-19
Revision Received:	2022-11-14
Editorial Decision:	2022-12-16
Revision Received:	2022-12-18
Accepted:	2022-12-19

Scientific Editor: Novella Guidi

Transaction Report:

September 19, 2022

Re: Life Science Alliance manuscript #LSA-2022-01614-T

Prof. Xilan Yu
Hubei University
College of Life Sciences
368 Youyi Avenue
Wuhan, Hubei 430062
China

Dear Dr. Yu,

Thank you for submitting your manuscript entitled "Regulation of telomere silencing by the core histones-autophagy-Sir2 axis during aging" to Life Science Alliance. The manuscript was assessed by expert reviewers, whose comments are appended to this letter. We invite you to submit a revised manuscript addressing the Reviewer comments.

Thank you for this interesting contribution to Life Science Alliance. We are looking forward to receiving your revised manuscript.

Sincerely,

B. MANUSCRIPT ORGANIZATION AND FORMATTING:

Reviewer #1 (Comments to the Authors (Required)):

In this manuscript, the authors have presented data that link the decrease in histone protein levels to the loss of telomeric silencing through the degradation of the Sir2 heterochromatic gene silencer by autophagy. The link between telomeric silencing, Sir2 levels and its regulation by the autophagic pathway was recently published in a paper by Zhang et al. in Nature Comm., which included a contribution by the senior author, Xilan Yu. Many of the same elegant genetic, molecular, and cellular tools to identify the silencing-Sir2-autophagy axis in this paper were employed in the present manuscript. While the Zhang et al. paper explored the role of histone H3T11 phosphorylation in this axis, the present ms. investigated whether histone protein levels also act in this same pathway. Using a series of mutants with decreased or increased levels of histones H3/H4, they provided solid and convincing evidence that this is the case, with statistics appropriately employed to reach their conclusions.

The ms. is well written and the conclusion that histone protein levels regulate telomeric silencing through adjusting Sir2 levels via autophagy adds interesting new information to this area of investigation. I have several comments, however, that relate to the authors' connection of their results to chronological aging. First, the majority of their studies were performed in proliferating cells, not in cells undergoing chronological aging. Second, their definition of chronological aging in yeast is not a standard one. Chronological aging is related to the proportion of quiescent cells present in a stationary phase (SP) culture and their survival over an extended period of time, generally measured as the proliferative capacity of cells over a period of weeks. The time frame that the authors describe as chronological aging (0-3 days or 0-6 days) is the period in which quiescent cells are formed. During this period, cells develop into non-quiescent (NQ) and quiescent (Q) cell populations, which can be isolated by density gradient centrifugation. Thus, in Figure 1, the authors need to repeat their investigation of histone levels not just by interrogating the mixed population of stationary phase cells but also the levels present in isolated NQ and Q cells. The loss of histones that they observe may occur predominantly in NQ cells, as these are budded cells that rapidly lose viability compared to Q cells, which are unbudded and stable. This separation should also be performed to measure H2A-td-tomato levels by microscopy. If it is found that histone loss is seen mostly in NQ cells, then it is likely that the silencing effects they observe occur because of the presence of NQ cells in their SP cultures at 3 and 6 days. While the results support the connection of histone levels to the silencing-Sir2-autophagy axis in proliferating cells, the Sir2-autophagy axis was not measured in their definition of chronologically aged cells, contrary to statements in the ms. Thus, I feel that the ms. title and the conclusions regarding chronological lifespan need to be re-visited.

Reviewer #2 (Comments to the Authors (Required)):

Review of LSA-2022-01614-T "Regulation of telomere silencing by the core histones-autophagy-Sir2 axis during aging" by Qianyun Mei, Qi Yu, Xin Li

Summary:

This manuscript describes studies in the budding yeast *Saccharomyces cerevisiae* on the regulation of silencing and histone density in cells undergoing a chronological aging assay, essentially cells grown to stationary phase in a flask and monitored during first week after growth arrest (often due to exhaustion of glucose in the medium). The authors monitor silencing by spot test and mRNA levels of genes in "heterochromatin" (genes 4 to 15 kb from telomeres), histone H3 and H4 levels by western analysis and autophagy by GFP-Atg8 degradation. The authors find that as H3 and H4 levels decrease during chronological aging, H3-H4 depletion is localized to telomere-adjacent genes and correlates with a decrease in Sir2 and silencing levels. Transcription of Sir2 does not change markedly in the three aging timepoints studied, but Sir2 protein levels decrease. The decrease appears to be due to autophagy as judged by Sir2-GFP seen exported from the nucleus and increases in Sir2 protein levels in a mutant defective for autophagy. Of note is the authors screening of the histone H3 and H4 point mutation library to identify amino acids that may modify Sir2 protein levels, telomere silencing, bulk H3/H4 levels and H3 levels at the telomere by ChIP. The histone mutants are sorted into different effects based on levels of silencing, GFP-Atg8 cleavage, bulk histone levels and H3-specific ChIP assays. In addition to these data, the authors claim to have "unveiled an epigenetic mechanism for how Set1-catalyzed H3K4me3 and its regulatory histone PTMs regulated telomere silencing."

Critique:

The authors present interesting, novel data in a crowded, established field. However, I have major reservations about some of the conclusions, experimental controls and confusing figures that need to be addressed.

1. A major issue is the term "core histones" used throughout the manuscript, which is an undefined jargon that can be interpreted multiple ways. As only one figure examines H2A levels, H2B is never examined and almost all of the work is on H3 and H4 forms, it is not clear what "core histones" means. Some readers would think "core" means the octamer to which many other proteins bind, including Sir2. Other might think it is the (H3-H4)₂ tetramer core to which H2A and H2B are attached. The term "core histones" is not well supported (or even defined) in the manuscript. The manuscript requires a rewrite that substitutes "core histones" with what the authors are actually assaying in each part of the manuscript, e.g. H3 or H4 when these proteins are specifically tested.

2. A second, potential major concern is that if histone H3 and H4 levels at telomeres decrease, the Sir2 protein has lost binding sites so excess Sir2 is present in cells. One could argue that degradation of "excess" Sir2 by autophagy is therefore not surprising. However, given that the authors have done a lot of work to move beyond the initial observation of autophagy, the manuscript does present data that can be of value to the field if other major concerns with controls and the writing can be addressed.

3. I have a major objection to the measure of autophagy by the ratio of (free GFP/GFP-Atg8), as this ratio is non-linear and overemphasizes the free GFP. This objection means that I do not believe that a role for H3K4A, H3K14A, H3R49A or H4R55A in autophagy has been shown (Fig. 5G). Briefly, in terms of the time frame of the GFP-Atg8 assay, the total level of GFP-Atg8 and free GFP is close to a constant. As GFP-Atg8 is degraded to free GFP, the GFP-Atg8 in the denominator will decrease and give a higher number for free GFP/GFP-Atg8. A graph would resemble a plot of $y = (x/(C-x))$ where x is bounded by 0 to C, and is non-linear with respect to autophagy levels. Regardless of whether this representation is a standard in the autophagy field, it unfairly skews the data. An analysis of free GFP/(GFP-Atg8 + free GFP) has to be done to judge the significance of the authors' claims. I cannot support publication without this analysis. The free GFP/GFP-Atg8 ratio is not a fair measure.

4. The conclusions are frequently too strong for the data presented. For example, H3K4Me3 is never directly assayed by ChIP or western, only by surrogates such as mutations in the methylation enzymes so changes in H3K4Me3 levels are only inferred. In addition, many of the histone mutations or changes during aging have the potential for other physiological changes that could impact the authors' assays. The conclusions in the Results and Discussion need to be tempered as many of the results are strongly suggestive or suggestive but are not proof. The statements should present the likely conclusions but allow for alternative hypotheses.

5. The anti-Sir2 Ab made by the authors was validated for western blotting but not for ChIP, which undercuts the ChIP results. The authors need to present controls that show that other Sir2-associated loci are in the ChIP (HMR, HML, rDNA) and that these levels are significantly decreased ChIPs from a sir2-deletion strain. Antibodies that work in westerns do not always give the same results in ChIP, so the support for the authors' claims of Sir2 occupancy near telomeres is not well-supported.

6. The Chi-square test is used throughout the manuscript but is not explained well enough in the Methods. An addition describing the degrees of freedom and how the comparisons are done is essential to judge the authors' statements of significance. The description on page 24 is too general, and numbers for the few reporter genes and telomeres assayed should be provided. As written, the analysis is not repeatable by other readers.

7. X and Y' elements near telomeres that affect the distance between the genes being assayed and the telomere are never mentioned or examined in the manuscript. Mutations in chromatin could impact how these elements move between telomeres in the many different strains used by the authors, or may have moved as the strains were propagated. For example, a Y' element added to a telomere between SOR1 or SOR2 in a histone mutant could alter the spread of silent chromatin or histone modifications from the telomere towards the centromere. Shampay and Blackburn 1988 had an enzymatic digestion for Southern blots that separated many individual telomeres with Y' elements that would allow one to show that the telomeres in the different mutants of interest in the manuscript do not have changes in the structure of the telomere that could have downstream effects on the assays. A PCR approach to test the individual telomeres nearest the reporter genes of interest may also be possible. The authors need to show that the telomeres nearest the reporter genes have the same distance between the reporter genes and the telomere repeats in the strains identified in the figures.

8. The references in the text are very odd, with the first citation showing the last author, then the first author et al. and year. Is this a Life Science Alliance standard? It makes the manuscript harder to read.

Specific comments:

Line 76: Which mechanism is being referred to, H3K4 methylation, modification in different subchromosomal domains or

something else?

Line 147: The low copy number plasmids used in this manuscript have to be described in this manuscript. Are they pRS-based YCp plasmids that are known to have copy numbers of 5 to 6 per cell? The authors need to define the plasmids and how the copy number is known. It is likely that the H3/H4 knockdown strains are a mix of cells without an H3/H4 divergent gene pair and some with more copies. While the western blots are reasonable, these details are important for interpreting the work.

Lines 180-184: The conclusions are too broad and in conflict with the data. The telomere-proximal genes in *hir1*-deletion mutants are not all significantly reduced.

Lines 616-617: Finish the reference.

Figures S4E and S4G: These expression plots have no meaning as there are not genes associated with the plots. Where are the telomere-proximal genes in these plots? Where are the ATG genes? My printout has no labels, only color. The main text and supplemental figure legend do not give enough description to make any point at all.

REVIEWER COMMENTS

Our responses are in blue.

Reviewer #1 (Comments to the Authors (Required)):

In this manuscript, the authors have presented data that link the decrease in histone protein levels to the loss of telomeric silencing through the degradation of the Sir2 heterochromatic gene silencer by autophagy. The link between telomeric silencing, Sir2 levels and its regulation by the autophagic pathway was recently published in a paper by Zhang et al. in Nature Comm., which included a contribution by the senior author, Xilan Yu. Many of the same elegant genetic, molecular, and cellular tools to identify the silencing-Sir2-autophagy axis in this paper were employed in the present manuscript. While the Zhang et al. paper explored the role of histone H3T11 phosphorylation in this axis, the present ms. investigated whether histone protein levels also act in this same pathway. Using a series of mutants with decreased or increased levels of histones H3/H4, they provided solid and convincing evidence that this is the case, with statistics appropriately employed to reach their conclusions.

The ms. is well written and the conclusion that histone protein levels regulate telomeric silencing through adjusting Sir2 levels via autophagy adds interesting new information to this area of investigation. I have several comments, however, that relate to the authors' connection of their results to chronological aging. First, the majority of their studies were performed in proliferating cells, not in cells undergoing chronological aging. Second, their definition of chronological aging in yeast is not a standard one. Chronological aging is related to the proportion of quiescent cells present in a stationary phase (SP) culture and their survival over an extended period of time, generally measured as the proliferative capacity of cells over a period of weeks. The time frame that the authors describe as chronological aging (0-3 days or 0-6 days) is the period in which quiescent cells are formed. During this period, cells develop into non-quiescent (NQ) and quiescent (Q) cell populations, which can be isolated by density gradient centrifugation. Thus, in Figure 1, the authors need to repeat their investigation of histone levels not just by interrogating the mixed population of stationary phase cells but also the levels present in isolated NQ and Q cells. The loss of histones that they observe may occur predominantly in NQ cells, as these are budded cells that rapidly lose viability compared to Q cells, which are unbudded and stable. This separation should also be performed to measure H2A-td-tomato levels by microscopy. If it is found that histone loss is seen mostly in NQ cells, then it is likely that the silencing effects they observe occur because of the presence of NQ cells in their SP cultures at 3 and 6 days. While the results support the connection of histone levels to the silencing-Sir2-autophagy axis in proliferating cells, the Sir2-autophagy axis was not measured in their definition of chronologically aged cells, contrary to

statements in the ms. Thus, I feel that the ms. title and the conclusions regarding chronological lifespan need to be re-visited.

We would like to thank this reviewer for constructive comments and suggestions. To study the changes of protein expression during chronological aging, we followed the protocols as described (*PLoS Genet* 2014, 10(1): e1004095; *Microb Cell* 2016, 3(3):109 -119). Walter et al. reported that the viability of yeast cells was reduced to 80% on day 1, 70% on day 3 and 50% on day 6 (Fig. 1A; *PLoS Genet* 2014, 10(1):e1004095). This suggests that the timeframe we studied (0-6 days) is chronological aging. We agree with the reviewer that during our study time period (0-6 days), cells may develop into non-quiescent (NQ) and quiescent (Q) cell populations (*Subcell Biochem* 2012, 57:123-43). In the revised manuscript, we isolated NQ and Q cells by density-gradient centrifugation according to the reviewer's suggestions. By immunoblots of cell lysates, we observed reduced four core histones in both NQ cells and Q cells (Fig. 1A). By microscopy, we observed reduced H2A-td-tomato levels in both NQ cells and Q cells (Fig. 1B-D). These results suggest that the four core histones are gradually lost in both NQ and Q cells. Therefore, our study is not only limited to proliferating cells but also chronologically aged cells. Nevertheless, to prevent confusion and make our description more accurate, we rephrased some sentences. The following are our changes (please see the manuscript with track of changes):

Line 34: Changed “regulate telomere silencing by modulating the core histones-autophagy-Sir2 axis during chronological aging” to “regulate telomere silencing by modulating the core histones-autophagy-Sir2 axis”.

Line 113: Changed “we grew yeast cells in YPD medium, which were aged for 0, 1, 3, 6 days.” to “we grew yeast cells in YPD medium for 0, 1, 3, 6 days.”

Line 114: Changed “By examining the histone protein levels in aged cells” to “By examining the histone protein levels in cells in stationary phase”.

Line 116: Changed “especially when cells were aged for 3 days and 6 days” to “especially when cells were grown for 3 days and 6 days”.

Line 119: Added “Yeast cells in stationary phase cultures usually develop into quiescent (Q) cells and non-quiescent (NQ) cells, which can be separated by density-gradient centrifugation (Allen et al, 2006). For both NQ and Q cells, we observed a gradual loss of four core histones (Fig. 1A).”

Line 123: Changed “This aged-coupled histone loss” to “This histone loss in the stationary phase cells”.

Line 125: Changed “We also isolated the cytoplasm and chromatin fractions from

cells that were chronologically aged for 0 and 3 days.” to “We also isolated the cytoplasm and chromatin fractions from cells that were grown for 0 and 3 days.”

Line 128: Changed “but not histone H2A variant H2AZ were reduced during aging” to “but not histone H2A variant H2AZ were reduced in stationary phase cells”.

Line 132: Changed “When cells were chronological aged for 0, 1, 3 and 6 days,” to “When cells were grown for 0, 1, 3 and 6 days”.

Line 136: Deleted “during chronological aging”.

Reviewer #2 (Comments to the Authors (Required)):

Review of LSA-2022-01614-T "Regulation of telomere silencing by the core histones-autophagy-Sir2 axis during aging" by Qianyun Mei, Qi Yu, Xin Li

Summary:

This manuscript describes studies in the budding yeast *Saccharomyces cerevisiae* on the regulation of silencing and histone density in cells undergoing a chronological aging assay, essentially cells grown to stationary phase in a flask and monitored during first week after growth arrest (often due to exhaustion of glucose in the medium). The authors monitor silencing by spot test and mRNA levels of genes in "heterochromatin" (genes 4 to 15 kb from telomeres), histone H3 and H4 levels by western analysis and autophagy by GFP-Atg8 degradation. The authors find that as H3 and H4 levels decrease during chronological aging, H3-H4 depletion is localized to telomere-adjacent genes and correlates with a decrease in Sir2 and silencing levels. Transcription of Sir2 does not change markedly in the three aging timepoints studied, but Sir2 protein levels decrease. The decrease appears to be due to autophagy as judged by Sir2-GFP seen exported from the nucleus and increases in Sir2 protein levels in a mutant defective for autophagy. Of note is the authors screening of the histone H3 and H4 point mutation library to identify amino acids that may modify Sir2 protein levels, telomere silencing, bulk H3/H4 levels and H3 levels at the telomere by ChIP. The histone mutants are sorted into different effects based on levels of silencing, GFP-Atg8 cleavage, bulk histone levels and H3-specific ChIP assays. In addition to these data, the authors claim to have "unveiled an epigenetic mechanism for how Set1-catalyzed H3K4me3 and its regulatory histone PTMs regulated telomere silencing."

Critique:

The authors present interesting, novel data in a crowded, established field. However, I have major reservations about some of the conclusions, experimental controls and

confusing figures that need to be addressed.

We would like to thank this reviewer for recognition of our work. We also thank this reviewer for constructive comments and suggestions. In the revised manuscript, we have addressed all the major concerns raised by the reviewer about some conclusions, controls and clarified figures as requested. The point-by-point answers to all the issues raised by the reviewer are enumerated below.

1. A major issue is the term “core histones” used throughout the manuscript, which is an undefined jargon that can be interpreted multiple ways. As only one figure examines H2A levels, H2B is never examined and almost all of the work is on H3 and H4 forms, it is not clear what "core histones" means. Some readers would think "core" means the octamer to which many other proteins bind, including Sir2. Other might think it is the (H3-H4)₂ tetramer core to which H2A and H2B are attached. The term "core histones" is not well supported (or even defined) in the manuscript. The manuscript requires a rewrite that substitutes "core histones" with what the authors are actually assaying in each part of the manuscript, e.g. H3 or H4 when these proteins are specifically tested.

We thank this reviewer for this point. We defined “core histones” as four canonical histones, including H2A, H2B, H3 and H4 in the revised manuscript. We also examined the amount of four core histones in Fig. 1A, 2D, 2F, 7A, 7D and 7F. For other results that have no H2A and H2B data, we rephrased the sentence to specify histones H3 and H4 instead of using “core histones”.

Line 79: Changed “The core histone genes are repressed” to “The transcription of four core histones (H2A, H2B, H3, H4) is repressed”.

Line 151: Changed “where two genomic copies of H3 and H4 are deleted and only one copy of histone H3 and H4 genes are expressed on a low copy number plasmid (Fig. 2A).” to “where two genomic copies of H3 and H4 are deleted and only one copy of histone H3 and H4 genes are expressed on a pRS-based low copy number plasmid (2-5 copies per cell) (Karim et al, 2013). The expression of histone H3 and H4 was reduced in H3/H4 KD cells (Fig. 2A)”.

Line 156: Deleted “, indicating that decreasing core histone levels reduces telomere silencing”

Line 157: Changed “To further show the relationship between core histone levels and telomere silencing, we constructed a plasmid that overexpresses histones H3 and H4 from a galactose-inducible” to “We also constructed a plasmid that overexpresses histones H3 and H4 from a galactose-inducible”.

Line 159: Changed “Cells transformed with the pGAL H3/H4 plasmid (H3/H4 OE) had more histones than cells transformed with empty vector” to “Cells transformed with the pGAL H3/H4 plasmid (H3/H4 OE) had more histone H3 and H4 than cells transformed with empty vector”.

Line 198: Changed “Overexpression of histones” to “Overexpression of histones H3 and H4”.

Line 226: Deleted “, suggesting that core histones protect Sir2 from being degraded by the vacuole-related autophagy pathway”.

Line 242: Changed “suggesting that reducing core histones” to “suggesting that reducing histones H3 and H4 levels”.

Line 271: Changed “knockdown of histones” to “knockdown of histones H3 and H4”.

Line 290: Changed “The core histone proteins” to “Histones H3 and H4”.

Line 357: Changed “by overexpression of core histones (Fig. 6G). Overexpression of core histones” to “by overexpression of core histones H3 and H4 (Fig. 6G). Overexpression of core histones H3 and H4”.

Line 363: Changed “histones” to “histones H3 and H4”.

Lines 831, 835, 843, 857, 859, 890: Added “(H3, H4)” or “H3 and H4”.

2. A second, potential major concern is that if histone H3 and H4 levels at telomeres decrease, the Sir2 protein has lost binding sites so excess Sir2 is present in cells. One could argue that degradation of "excess" Sir2 by autophagy is therefore not surprising. However, given that the authors have done a lot of work to move beyond the initial observation of autophagy, the manuscript does present data that can be of value to the field if other major concerns with controls and the writing can be addressed.

We would like to thank this reviewer for constructive comments and suggestions. In the revised manuscript, we addressed all the concerns with controls, i.e., Sir2 antibody (please see our answer to question 5). We also carefully edited the whole manuscript to improve its quality.

3. I have a major objection to the measure of autophagy by the ratio of (free GFP/GFP-Atg8), as this ratio is non-linear and overemphasizes the free GFP. This objection means that I do not believe that a role for H3K4A, H3K14A, H3R49A or H4R55A in autophagy has been shown (Fig. 5G). Briefly, in terms of the time frame

of the GFP-Atg8 assay, the total level of GFP-Atg8 and free GFP is close to a constant. As GFP-Atg8 is degraded to free GFP, the GFP-Atg8 in the denominator will decrease and give a higher number for free GFP/GFP-Atg8. A graph would resemble a plot of $y = (x/(C-x))$ where x is bounded by 0 to C , and is non-linear with respect to autophagy levels. Regardless of whether this representation is a standard in the autophagy field, it unfairly skews the data. An analysis of free GFP/(GFP-Atg8 + free GFP) has to be done to judge the significance of the authors' claims. I cannot support publication without this analysis. The free GFP/GFP-Atg8 ratio is not a fair measure.

We thank this reviewer for this comment. As per requested by the reviewer, we analyzed the ratio of free GFP/(GFP-Atg8 + free GFP) for all autophagy activity in figures 4I, 5G, S4C. Our data still support our conclusion (Fig. 4I, 5G, S4C).

Line 264: Changed “free GFP/GFP-Atg8” to “free GFP/(free GFP+GFP-Atg8)”.

4. The conclusions are frequently too strong for the data presented. For example, H3K4Me3 is never directly assayed by ChIP or western, only by surrogates such as mutations in the methylation enzymes so changes in H3K4Me3 levels are only inferred. In addition, many of the histone mutations or changes during aging have the potential for other physiological changes that could impact the authors' assays. The conclusions in the Results and Discussion need to be tempered as many of the results are strongly suggestive or suggestive but are not proof. The statements should present the likely conclusions but allow for alternative hypotheses.

We would like to thank this reviewer for constructive comments and suggestions. In the revised manuscript, we tempered some conclusions to be more accurate. We also replaced “indicate” with “suggest” at some places. For H3K4me3, we performed ChIP-qPCR for H3K4me3 at histone genes in WT, H3K4A, *set1Δ* and *spp1Δ* mutants to show that Set1 catalyzes H3K4me3 at histone genes to promote histone gene expression. These data were presented in Fig. S7A, S7B and S7C.

Line 226: Deleted “, suggesting that core histones protect Sir2 from being degraded by the vacuole-related autophagy pathway”.

Line 256: Changed “All these data indicate” to “All these data suggest”.

Line 325: Changed “indicating” to “suggesting”.

Line 340: Changed “Mutation of H3K4A or loss of Set1 reduced the global levels of histone proteins as well as histone occupancy at telomere proximity genes but not euchromatic genes (Fig. 6A and B; Fig. S7A).” to “Mutation of H3K4A and loss of Set1 reduced the occupancy of H3K4me3 at histone genes and decreased the global levels of histone proteins (Fig. S7A-C). The histone occupancy at telomere-proximal

genes was also significantly reduced in H3K4A and *set1Δ* mutants (Fig. 6A and B).”

Line 345: Changed “(Fig. 6C, S7B).” to “(Fig. 6C, S7D and E).”.

Line 364: Deleted “, suggesting that Set1-catalyzed H3K4me3 promotes histone gene expression to maintain Sir2 homeostasis.”

Line 366: Deleted “, suggesting that Set1-catalyzed H3K4me3 prevents autophagy-mediated Sir2 degradation”

Line 369: Deleted “, indicating that Set1-catalyzed H3K4me3 inhibits the autophagy pathway”.

Line 395: Deleted “-catalyzed H3K4me3”.

5. The anti-Sir2 Ab made by the authors was validated for western blotting but not for ChIP, which undercuts the ChIP results. The authors need to present controls that show that other Sir2-associated loci are in the ChIP (HMR, HML, rDNA) and that these levels are significantly decreased ChIPs from a *sir2*-deletion strain. Antibodies that work in westerns do not always give the same results in ChIP, so the support for the authors' claims of Sir2 occupancy near telomeres is not well-supported.

We would like to thank this reviewer for constructive comments and suggestions. In the revised manuscript, we performed ChIP-qPCR for Sir2 binding at regions (telomere-proximal genes, HMR, HML, rDNA) using anti-Sir2 antibody in WT and *sir2Δ* mutant. Our data showed that there were significant signals for Sir2 binding at HMR, HML, rDNA in WT but not in *sir2Δ* mutant (Fig. S5I, bottom panel), suggesting that the anti-Sir2 antibody is specific for Sir2 and can be used for ChIP.

Fig. S5I figure legend: Added “The bottom panel: ChIP-qPCR analysis of Sir2 occupancy at telomere-proximal genes, rDNA, HML and HMR in WT and *sir2Δ* mutant using anti-Sir2 antibody.”

6. The Chi-square test is used throughout the manuscript but is not explained well enough in the Methods. An addition describing the degrees of freedom and how the comparisons are done is essential to judge the authors' statements of significance. The description on page 24 is too general, and numbers for the few reporter genes and telomeres assayed should be provided. As written, the analysis is not repeatable by other readers.

We would like to thank this reviewer for constructive comments and suggestions. In the revised manuscript, we added the following sentences to describe how performed

the comparisons to judge the statement of significance: “The χ^2 values score the significance of the difference between the expected frequency from the genome-wide average to the observed frequency in each 10 kb interval with the degree of freedom of 1”. In addition, we provided the numbers of the genes and telomeres in the Supplementary Table S4.

Line 622: Added “The χ^2 values score the significance of the difference between the expected frequency from the genome-wide average to the observed frequency in each 10 kb interval with the degree of freedom of 1”.

Line 626: Added “and listed in Supplementary Table S4”.

7. X and Y’ elements near telomeres that affect the distance between the genes being assayed and the telomere are never mentioned or examined in the manuscript. Mutations in chromatin could impact how these elements move between telomeres in the many different strains used by the authors, or may have moved as the strains were propagated. For example, a Y’ element added to a telomere between SOR1 or SOR2 in a histone mutant could alter the spread of silent chromatin or histone modifications from the telomere towards the centromere. Shampay and Blackburn 1988 had an enzymatic digestion for Southern blots that separated many individual telomeres with Y’ elements that would allow one to show that the telomeres in the different mutants of interest in the manuscript do not have changes in the structure of the telomere that could have downstream effects on the assays. A PCR approach to test the individual telomeres nearest the reporter genes of interest may also be possible. The authors need to show that the telomeres nearest the reporter genes have the same distance between the reporter genes and the telomere repeats in the strains identified in the figures.

That is a good point. To examine the effect of histone mutations and histone regulators on telomere length, we performed Southern blots to detect Y’ and terminal telomere fragments using the method developed by Shampay and Blackburn. Our data showed that overexpression or knockdown of histones H3/H4 had no effect on telomeric fragment (Fig. S8E). In addition, the telomere length in *spt21 Δ* , *hir1 Δ* , *hir2 Δ* , *hir3 Δ* , *set1 Δ* and *spp1 Δ* was indistinguishable from WT (Fig. S8E). For histone mutations (H3R2A, H3K4A, H3T6A, H3K14A, H3R40A, H3R49A, H3K56A, H3R72A, H3D77A, H3Q85A, H4L37A, H4K44A, H4R55A), they had similar telomere length with their WT counterpart (Fig. S8F). All these data suggest that these proteins and histone mutants regulate telomere silencing not by affecting the telomere length and Y’ repeats.

Line 511: Added “Yeast telomeres contain Y’ repeats and variable X sequences, which may affect the distance between the genes being assayed and the nearest telomere and eventually influence their silencing. We thus performed Southern blots to detect Y’ and terminal telomere fragments in the mutants examined in this study

(Shampay and Blackburn, 1988; Tsukamoto et al, 2005). Cells were collected and DNA was cut with XhoI and probed with a Y'-TG probe that gave three fragments, 6.5, 5.5, and 1.3 kb in WT cells. The two larger fragments correspond to repetitive Y', while the shortest fragment is the terminal 1 kb of Y' and also contains ~350 bp of telomeric TG repeats. Our data showed that overexpression or knockdown of histones H3/H4 had no effect on telomeric fragment (Fig. S8E). In addition, the telomere length in *spt21Δ*, *hir1Δ*, *hir2Δ*, *hir3Δ*, *set1Δ* and *spp1Δ* was indistinguishable from WT (Fig. S8E). For histone mutations, H3R2A, H3K4A, H3T6A, H3K14A, H3R40A, H3R49A, H3K56A, H3R72A, H3D77A, H3Q85A, H4L37A, H4K44A and H4R55A had similar telomere length with their WT counterpart (Fig. S8F). These data suggest that these proteins or histone mutants regulate telomere silencing not by affecting the telomere length and Y' repeats."

8. The references in the text are very odd, with the first citation showing the last author, then the first author et al. and year. Is this a Life Science Alliance standard? It makes the manuscript harder to read.

We would like to thank this reviewer for constructive comments and suggestions. We carefully edited all references according to Life Science Alliance standard.

Specific comments:

1. Line 76: Which mechanism is being referred to, H3K4 methylation, modification in different subchromosomal domains or something else?

Line 74: Changed "however, the underlying mechanism remains largely unknown." to "however, the underlying mechanism by which H3K4me3 regulates telomere silencing remains largely unknown."

2. Line 147: The low copy number plasmids used in this manuscript have to be described in this manuscript. Are they pRS-based YCp plasmids that are known to have copy numbers of 5 to 6 per cell? The authors need to define the plasmids and how the copy number is known. It is likely that the H3/H4 knockdown strains are a mix of cells without an H3/H4 divergent gene pair and some with more copies. While the western blots are reasonable, these details are important for interpreting the work.

We thank this reviewer for this comment. The plasmid used in this work was pRS-based low copy number plasmid, which has been reported to have 2-5 copies per cell (*FEMS Yeast Res* 2013, 13(1):107-16). We added this information in the result section. We cannot exclude the possibility that the H3/H4 knockdown strain is a mix of cells without an H3/H4 divergent gene pair and some with more copies. As our

Western blots already confirmed that histones H3 and H4 were reduced in this mutant, we named this strain as H3/H4 KD cells.

Line 151: Changed “where two genomic copies of H3 and H4 are deleted and only one copy of histone H3 and H4 genes are expressed on a low copy number plasmid (Fig. 2A).” to “where two genomic copies of H3 and H4 are deleted and only one copy of histone H3 and H4 genes are expressed on a pRS-based low copy number plasmid (2-5 copies per cell) (Karim et al, 2013). The expression of histone H3 and H4 was reduced in H3/H4 KD cells (Fig. 2A)”.

3. Lines 180-184: The conclusions are too broad and in conflict with the data. The telomere-proximal genes in *hir1*-deletion mutants are not all significantly reduced.

To make this conclusion more accurate, we changed “The transcription of telomere-proximal genes but not euchromatic genes was significantly reduced in *hir1Δ*, *hir2Δ* and *hir3Δ* mutants when compared to WT (Fig. 2G)” to “The transcription of telomere-proximal genes but not euchromatic genes was significantly reduced in *hir2Δ* and *hir3Δ* mutants (Fig. 2F). The transcription of *PHO11* and *YDL241W* was significantly reduced in *hir1Δ* (Fig. 2F). Collectively, these data suggest that intracellular core histones tightly control telomere silencing.” (Line 188)

4. Lines 616-617: Finish the reference.

We finished the reference as “Chakravarti D, LaBella KA, DePinho RA. 2021. Telomeres: History, health, and hallmarks of aging. *Cell* 184(2):306-322. doi:10.1016/j.cell.2020.12.028” (Line 664).

5. Figures S4E and S4G: These expression plots have no meaning as there are not genes associated with the plots. Where are the telomere-proximal genes in these plots? Where are the ATG genes? My printout has no labels, only color. The main text and supplemental figure legend do not give enough description to make any point at all.

Figures S4E and S4G showed the transcription changes of autophagy genes in *spt21Δ*, *hir1Δ*, *hir2Δ* and *hir3Δ* mutants. To make this clearer, we added the labels for ATG genes in Figures S4E and S4G in the revised manuscript.

December 16, 2022

RE: Life Science Alliance Manuscript #LSA-2022-01614-TR

Prof. Xilan Yu
Hubei University
College of Life Sciences
368 Youyi Avenue
Wuhan, Hubei 430062
China

Dear Dr. Yu,

Thank you for submitting your revised manuscript entitled "Regulation of telomere silencing by the core histones-autophagy-Sir2 axis during aging". We would be happy to publish your paper in Life Science Alliance pending final revisions necessary to meet our formatting guidelines.

- please address carefully the remaining Reviewer 1 and 2 points
- please upload your supplemental figures as single files and add your supplemental figure legends to the main manuscript text
- please add the Twitter handle of your host institute/organization as well as your own or/and one of the authors in our system
- please upload your tables as editable doc or excel files or make sure the table files are in the doc file of your manuscript; please add the table legends to the main figure legends section

Figure Check:

- Figure 1A Middle section (NQ Cells), 2nd row: there is a thick white line in the column label 3. Please provide source data
- Figure 1E, bottom row, blot in 3rd column looks as if it's been pasted in. Please provide source data

A. FINAL FILES:

B. MANUSCRIPT ORGANIZATION AND FORMATTING:

Sincerely,

Reviewer #1 (Comments to the Authors (Required)):

In this revised ms., the authors extensively addressed the main points related to their conclusion that telomeric silencing is regulated by the levels of the core histones, which in turn regulate the levels of the Sir2 silencing factor via autophagy. I found the results interesting and the experiments well-done and nicely controlled, and think that the authors in general successfully addressed the major criticisms raised by reviewer 2. However, I am still troubled by the authors' title, which specifically relates their findings to events during chronological aging. The majority of their studies were performed in exponential cells. What they refer to as chronologically aging is very misleading. For most of their aging experiments, they used cells collected at 0 time, presumably exponentially growing cells, and 3 day post-inoculation, 6-day post inoculation, and occasionally 1-day post inoculation cells. Based on studies performed in many labs, 1-day post-inoculation cells usually represent cells that have exhausted their carbon sources and begun their development into quiescent (Q)and non-quiescent (NQ) cells. Three-day post-inoculation usually represents the cell doubling that completes production of Q and NQ cells, which then further mature by 5-6 days after inoculation, or stationary phase. Chronological lifespan or aging is regulated by the fraction and stability of Q cells in a stationary phase culture over time, most often assayed days or weeks after stationary phase is achieved. While one might argue that chronological aging is initiated when Q and NQ cells are formed at day 3, it is not standard to consider it only 3 days later, or at 6-days post-inoculation. The authors presented only one standard CLS assay in the supplement (S8A-C). My suggestion is the following. Remove the term aging from the title; move Figure 1 to later in the ms., before Fig. 7, and include Fig.S1 (replicative aging) as a supplement to these two figures. The discussion could then be framed to ask if the results seen in growing cells apply to cells that have "initiated" chronological aging, as is apparently the case. I think the authors need to be careful in their definition of chronological aging, and doing so should not distract from their overall nice study.

Reviewer #2 (Comments to the Authors (Required)):

Review of revised submission LSA-2022-01614-TR "Regulation of telomere silencing by the core histones-autophagy-Sir2 axis during aging" by Qianyun Mei, Qi Yu, Xin Li, Jianguo Chen, Xilan Yu

In this revised submission, the authors have responded to almost all of my comments in a substantial manner. The English of the manuscript is markedly improved. The conclusions are more closely related to the data presented and not going beyond what is justified. Additional experiments are now included to address specific comments that caused great concern to this reviewer in the first submission. Overall, the manuscript is markedly improved, but has one concern worth noting that can be

easily fixed.

In my previous comment number 7, I raised the concern that the distance between the very end of the chromosome and the reporter may change because Y' elements are mobile and can recombine to form tandem Y' elements at some telomeres. In this case, a telomere with one Y' element could end up with 2 to 4 Y' elements. A case like this one was described in PubMed ID 8319906 in Figures 2 and 3, which increases the distance between the very terminal TG1-3 repeats at the chromosome end and the reporter gene. I suggested the Shampay and Blackburn 1988 paper as way to analyze the size of the telomeres with their reporter genes of interest because they used a Pvu II digest to look at individual telomeres, which could show if the length of some telomeres were changing by the insertion of 5.5 or 6.7 kb Y' element. However, I did not have access to the 1988 paper when I finished the review and could not cite the restriction enzyme to be used or mention that the probe should be telomere repeats, not Y' hybridizing DNA.

The authors instead used Xho I digests to look at the length of the terminal TG1-3 repeats and amount of tandem Y' elements in Figure S8. While no gross changes are detected, it does not answer the original question and the statements in the manuscript go beyond the data. However, given the amount of work the authors have already done, it seems reasonable to rewrite a few sentences related to the data they present instead of directly addressing my original question. The focus should be on the length of the terminal repeat tracts and the number of Y' elements, which the authors assayed. I suggest the following:

Page 20, line 491: change "Yeast telomeres contain Y' repeats and variable X sequences, which may affect the" to "Yeast telomeres contain Y' repeats and variable X sequences, the numbers of which may affect the"

Page 21, lines 503-505: change "These data suggest that these proteins or histone mutants regulate telomere silencing not by affecting the telomere length and Y' repeats." to

"These data suggest that the effect of these protein or hisone mutants on telomere silencing is not due to major changes in length of the terminal telomere repeat tract or the number of Y' repeats."

With these small changes, I have no other objections to the manuscript.

REVIEWER COMMENTS

Our responses are in blue.

Reviewer #1 (Comments to the Authors (Required)):

In this revised ms., the authors extensively addressed the main points related to their conclusion that telomeric silencing is regulated by the levels of the core histones, which in turn regulate the levels of the Sir2 silencing factor via autophagy. I found the results interesting and the experiments well-done and nicely controlled, and think that the authors in general successfully addressed the major criticisms raised by reviewer 2. However, I am still troubled by the authors' title, which specifically relates their findings to events during chronological aging. The majority of their studies were performed in exponential cells. What they refer to as chronologically aging is very misleading. For most of their aging experiments, they used cells collected at 0 time, presumably exponentially growing cells, and 3 day post-inoculation, 6-day post inoculation, and occasionally 1-day post inoculation cells. Based on studies performed in many labs, 1-day post-inoculation cells usually represent cells that have exhausted their carbon sources and begun their development into quiescent (Q) and non-quiescent (NQ) cells. Three-day post-inoculation usually represents the cell doubling that completes production of Q and NQ cells, which then further mature by 5-6 days after inoculation, or stationary phase. Chronological lifespan or aging is regulated by the fraction and stability of Q cells in a stationary phase culture over time, most often assayed days or weeks after stationary phase is achieved. While one might argue that chronological aging is initiated when Q and NQ cells are formed at day 3, it is not standard to consider it only 3 days later, or at 6-days post-inoculation. The authors presented only one standard CLS assay in the supplement (S8A-C).

My suggestion is the following. Remove the term aging from the title; move Figure 1 to later in the ms., before Fig. 7, and include Fig.S1 (replicative aging) as a supplement to these two figures. The discussion could then be framed to ask if the results seen in growing cells apply to cells that have "initiated" chronological aging, as is apparently the case. I think the authors need to be careful in their definition of chronological aging, and doing so should not distract from their overall nice study.

We would like to thank this reviewer for constructive suggestions. We made the following changes as suggested by the reviewer:

1. We removed the “aging” from the title.
2. We moved Figure 1 to Figure 6.
3. We moved Fig. S1 to Fig. S7 as a supplement to Fig. 6 and Fig. 7.
4. We changed “chronological aging” to “have initiated chronological aging” in the revised manuscript wherever possible.

5. We also emphasized our story at growing cells.

Reviewer #2 (Comments to the Authors (Required)):

Review of revised submission LSA-2022-01614-TR "Regulation of telomere silencing by the core histones-autophagy-Sir2 axis during aging" by Qianyun Mei, Qi Yu, Xin Li, Jianguo Chen, Xilan Yu

In this revised submission, the authors have responded to almost all of my comments in a substantial manner. The English of the manuscript is markedly improved. The conclusions are more closely related to the data presented and not going beyond what is justified. Additional experiments are now included to address specific comments that caused great concern to this reviewer in the first submission. Overall, the manuscript is markedly improved, but has one concern worth noting that can be easily fixed.

In my previous comment number 7, I raised the concern that the distance between the very end of the chromosome and the reporter may change because Y' elements are mobile and can recombine to form tandem Y' elements at some telomeres. In this case, a telomere with one Y' element could end up with 2 to 4 Y' elements. A case like this one was described in PubMed ID 8319906 in Figures 2 and 3, which increases the distance between the very terminal TG1-3 repeats at the chromosome end and the reporter gene. I suggested the Shampay and Blackburn 1988 paper as way to analyze the size of the telomeres with their reporter genes of interest because they used a Pvu II digest to look at individual telomeres, which could show if the length of some telomeres were changing by the insertion of 5.5 or 6.7 kb Y' element. However, I did not have access to the 1988 paper when I finished the review and could not cite the restriction enzyme to be used or mention that the probe should be telomere repeats, not Y' hybridizing DNA.

The authors instead used Xho I digests to look at the length of the terminal TG1-3 repeats and amount of tandem Y' elements in Figure S8. While no gross changes are detected, it does not answer the original question and the statements in the manuscript go beyond the data. However, given the amount of work the authors have already done, it seems reasonable to rewrite a few sentences related to the data they present instead of directly addressing my original question. The focus should be on the length of the terminal repeat tracts and the number of Y' elements, which the authors assayed. I suggest the following:

Page 20, line 491: change "Yeast telomeres contain Y' repeats and variable X sequences, which may affect the" to "Yeast telomeres contain Y' repeats and variable X sequences, the numbers of which may affect the"

Page 21, lines 503-505: change "These data suggest that these proteins or histone mutants regulate telomere silencing not by affecting the telomere length and Y' repeats." To

"These data suggest that the effect of these protein or histone mutants on telomere silencing is not due to major changes in length of the terminal telomere repeat tract or the number of Y' repeats."

With these small changes, I have no other objections to the manuscript.

We would like to thank this reviewer for constructive suggestions. We made the following changes as suggested by the reviewer:

1. Page 20, line 495: change "Yeast telomeres contain Y' repeats and variable X sequences, which may affect the" to "Yeast telomeres contain Y' repeats and variable X sequences, the numbers of which may affect the".
2. Page 21, lines 507: change "These data suggest that these proteins or histone mutants regulate telomere silencing not by affecting the telomere length and Y' repeats." To "These data suggest that the effect of these protein or histone mutants on telomere silencing is not due to major changes in length of the terminal telomere repeat tract or the number of Y' repeats."

December 19, 2022

RE: Life Science Alliance Manuscript #LSA-2022-01614-TRR

Prof. Xilan Yu
Hubei University
College of Life Sciences
368 Youyi Avenue
Wuhan, Hubei 430062
China

Dear Dr. Yu,

Thank you for submitting your Research Article entitled "Regulation of telomere silencing by the core histones-autophagy-Sir2 axis". It is a pleasure to let you know that your manuscript is now accepted for publication in Life Science Alliance. Congratulations on this interesting work.

DISTRIBUTION OF MATERIALS:

Again, congratulations on a very nice paper. I hope you found the review process to be constructive and are pleased with how the manuscript was handled editorially. We look forward to future exciting submissions from your lab.

Sincerely,
